# Hierarchical tensile structures with ultralow mechanical dissipation

M. J. Bereyhi [1,2], A. Beccari [1,2], R. Groth[1,2], S. A. Fedorov[1,2], A. Arabmoheghi[1], T. J. Kippenberg [1✉] & N. J. Engelsen [1✉]

Structural hierarchy is found in myriad biological systems and has improved man-made structures ranging from the Eiffel tower to optical cavities. In mechanical resonators whose rigidity is provided by static tension, structural hierarchy can reduce the dissipation of the fundamental mode to ultralow levels due to an unconventional form of soft clamping. Here, we apply hierarchical design to silicon nitride nanomechanical resonators and realize binary tree-shaped resonators with room temperature quality factors as high as $7.8 \times 10^8$ at 107 kHz frequency ($1.1 \times 10^9$ at $T = 6$ K). The resonators' thermal-noise-limited force sensitivities reach 740 zN/Hz$^{1/2}$ at room temperature and 90 zN/Hz$^{1/2}$ at 6 K, surpassing state-of-the-art cantilevers currently used for force microscopy. Moreover, we demonstrate hierarchically structured, ultralow dissipation membranes suitable for interferometric position measurements in Fabry-Pérot cavities. Hierarchical nanomechanical resonators open new avenues in force sensing, signal transduction and quantum optomechanics, where low dissipation is paramount and operation with the fundamental mode is often advantageous.

[1] Institute of Physics, Swiss Federal Institute of Technology Lausanne (EPFL), 1015 Lausanne, Switzerland. [2] These authors contributed equally: M. J. Bereyhi, A. Beccari, R. Groth, S. A. Fedorov. ✉email: tobias.kippenberg@epfl.ch; nils.engelsen@epfl.ch

Structural hierarchy increases mechanical stiffness in animal bones[1] and artificial network materials[2], reduces the weight of load-bearing structures[3], allows efficient delivery of air to alveoli in lungs[4] and of fluids in vascular systems[5], confines the optical field at a deeply sub-wavelength scale[6], and provides unique opportunities for the reduction of mechanical dissipation in strained materials[7]. When structural hierarchy is accompanied by self-similarity at different scales, it leads to fractal-like features. Self-similar mechanical resonators can have acoustic mode densities consistent with non-integer dimensionality[8], similar to the spectra of natural compounds with scale invariance, such as proteins[9], silica aerogels, and glasses[8].

Strained mechanical resonators can have ultralow dissipation owing to the effect of dissipation dilution[10,11], whereby the intrinsic material friction is diluted via tension. This phenomenon was first explored in the mirror suspensions of gravitational wave detectors[10] and over the past decade has been exploited to reduce dissipation at the nanoscale[12,13]. Dissipation dilution is strongly impacted by the resonator geometry, which offers a practical way of reducing mechanical losses. Substantial improvements have been achieved by phononic crystal-based soft-clamping[14] and elastic strain engineering[15], enabling amorphous nanomechanical devices to closely approach the quality factors (~$10^9$) of the lowest dissipation macroscopic oscillators, such as single-crystal quartz and sapphire bulk resonators[16,17]. However, these techniques apply only to high-order modes (~10 to 100) of strings and membranes, imposing experimental limitations in quantum optomechanics such as intermodulation noise[18,19] and instability of lower order mechanical modes[20]. Furthermore, phononic bandgap engineering is impractical at low frequencies due to the large device sizes required (tens of millimeters for the 100 kHz range). Contemporaneous work such as perimeter modes[21] and "spider-web" resonators[22] have demonstrated low dissipation at low frequencies, but not for the fundamental mode of the structure.

Here we employ structural hierarchy to realize an unconventional form of soft clamping for the fundamental mode[7]. These resonators are exquisite force sensors by virtue of their low loss, low mass, and low resonance frequency (50 kHz to 1 MHz in this work). Our resonators' vibrational excitations undergo very slow thermal decoherence due to their high quality factors (up to $Q = 7.8 \times 10^8$). At room temperature, the thermal decoherence rate, $\Gamma_d = k_B T/\hbar Q$[23] ($k_B$ is Boltzmann's constant, $\hbar$ is the reduced Planck's constant and $T$ is the temperature), of our best devices is below 10 kHz—comparable to the decoherence rates of dielectric nanoparticles trapped in a laser beam[24]. These properties make hierarchical resonators well-suited for sensing applications[25,26] and a multitude of quantum optomechanics experiments, such as ground-state cooling[20,23], microwave-optical photon conversion[27] and squeezing of an optical field[28].

The hierarchical principle is used in our resonators' connections to their chip frames (illustrated in Fig. 1a, b). The end members are ribbon segments with a rectangular cross section and are clamped at one end. These segments contribute a large portion of the internal friction, which increases with the gradient angle $\gamma$ at which the vibrational mode approaches the clamped boundary. If a simple end segment is replaced with a bifurcating junction as shown in Fig. 1a, the mode gradient near the clamping points is reduced, as shown in Fig. 1b. Provided the wavelength is much larger than all segment lengths, the new gradient $\gamma'$ is related to the original one as

$$\gamma' = \gamma \cos(\theta), \qquad (1)$$

where $\theta$ is the junction branching angle. Successive junctions reduce the mode gradient further and after a sufficient number of iterations soft clamping is achieved[7], as shown in Fig. 1c. The

reduced mode amplitude at the clamping points also results in suppression of acoustic radiation losses of the fundamental mode. We use hierarchical junctions to design several types of soft-clamped resonators: binary tree beams, trampoline membranes with branching tethers and "steering wheel" trampolines (see Fig. 1d–f). Among soft-clamped resonators, hierarchical resonators offer the largest relative frequency separation for the high-$Q$ mode (see Supplementary Information), which reduces the off-resonant thermal noise contribution from adjacent mechanical modes. In Fig. 1g, we make a comparison of the quality factors of our designs with the current state of the art, PnC soft clamping combined with strain engineering[11,15], subject to the same material and size constraints. The soft clamped modes of our devices have as high quality factors as theoretically allowed for PnC resonators, but their frequency range and mode order are inaccessibly low for PnC designs. Compared to a uniform, doubly-clamped beam fundamental mode with the same frequency, our devices can have quality factors more than one order of magnitude higher.

## Results

We first explore one-dimensional hierarchical mechanical resonators in the form of binary tree beams. Each binary tree resonator is defined by two self-similar trees joined at their 0th generation segments. The length of the 0th generation segment, $2l_0$, determines the fundamental mode frequency of the device. The length of segments in generation $k$ is set by multiplying $l_0$ with a factor of $(r_l)^k$, where the length contraction ratio $r_l < 1$ is chosen to be sufficiently small that no self-contact occurs. The widths of the segments are determined by requiring constant stress throughout the structure, which minimizes static in-plane deformations upon structure undercut for higher fabrication yield. This requirement is fulfilled by multiplying the width of each new generation of segments by a factor of $1/(2\cos\theta)$, which typically makes outer segments wider as is illustrated in the inset of Fig. 2a. In the resulting devices, the stress along all segments is equal to the film deposition stress[7]. The devices in our work are fabricated from suspended 20 nm-thick, high-stress stoichiometric silicon nitride ($Si_3N_4$) films grown on a silicon substrate; fabrication details are provided in the "Methods" section.

Figure 2 summarizes our study of the quality factors of binary tree resonators optimized for low dissipation. Figure 2a–c shows scanning electron microscope images of 20-nm-thick binary tree beams, with deposition stress $\sigma_{dep} = 1.0$ GPa and intrinsic quality factor $Q_{int} = 2500$ at room temperature (see "Methods"). These devices have a central segment length ranging between $2l_0 = 360$ μm and $2l_0 = 4$ mm (and total longitudinal extent up to 5.5 mm), branching angles between $\theta = 75°$ and $\theta = 83°$, and length contraction ratios between $r_l = 0.45$ and $r_l = 0.67$. The dissipation rates were characterized by ringdown measurements with optical interferometric position readout (see "Methods"). The probe laser beam was gated and switched on only for short periods to eliminate the possibility of photothermal damping (see Supplementary Information). While most of the data was collected at room temperature, some devices were also cooled down to 6 K in a cryostat. In the following, we refer to room temperature measurements unless otherwise specified. An example amplitude ringdown is shown in Fig. 2f for a 110 kHz-frequency mode with a time constant of 20 min (damping rate $\Gamma_m/2\pi = 140$ μHz); the mechanical signal remained above the shot noise of the laser probe for several hours. Introducing three branching generations enhances the fundamental mode $Q$ by a factor of about 30 beyond the value exhibited by a doubly-clamped beam of the same frequency. The measured quality factors of fundamental modes are summarized in Fig. 2e. Devices

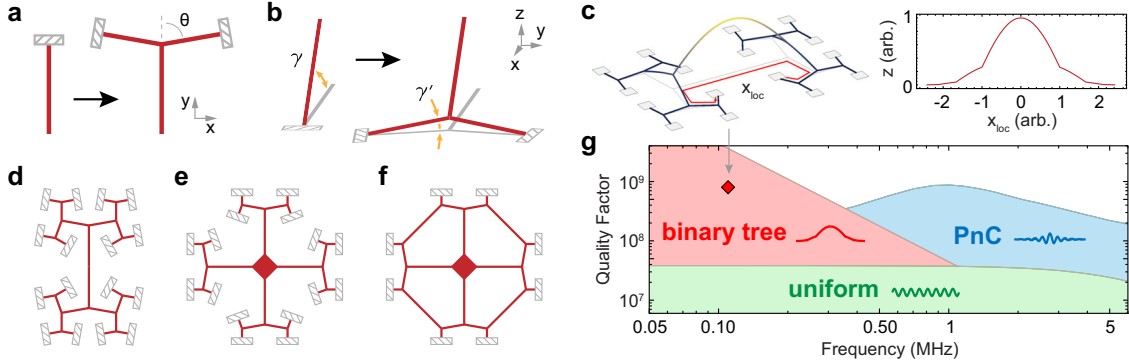

**Fig. 1 Construction of hierarchical nanomechanical resonators. a** The elementary substitution operation in the hierarchical junction approach and **b** the effect it produces on the deformation gradients when the structure is deformed out of the *xy* plane. Dashed rectangles indicate the supports to which the structure is clamped. **c** Displacement profile of a binary tree beam fundamental mode, displayed in 3D (left), and evaluated along the red path $x_{loc}$ (right). The abrupt mode gradient changes appear due to plotting the mode along a broken line that follows the hierarchy of the branches (red arrow path in the 3D model). As illustrated in the 3D mode profile the displacement undergoes a smooth transition towards the clamping points. Schematic depiction of nanomechanical resonator designs: a binary tree beam (**d**), a trampoline with branching tethers (**e**), and a steering wheel trampoline (**f**). **g** Diagram of dissipation dilution in stressed resonators. The top boundaries of colored regions indicate the theoretical $Q$ vs. frequency limits for different types of $Si_3N_4$ resonators with 20 nm thicknesses and lengths below 3 mm. The red diamond indicates the mode with the highest $Q$ measured in this work.

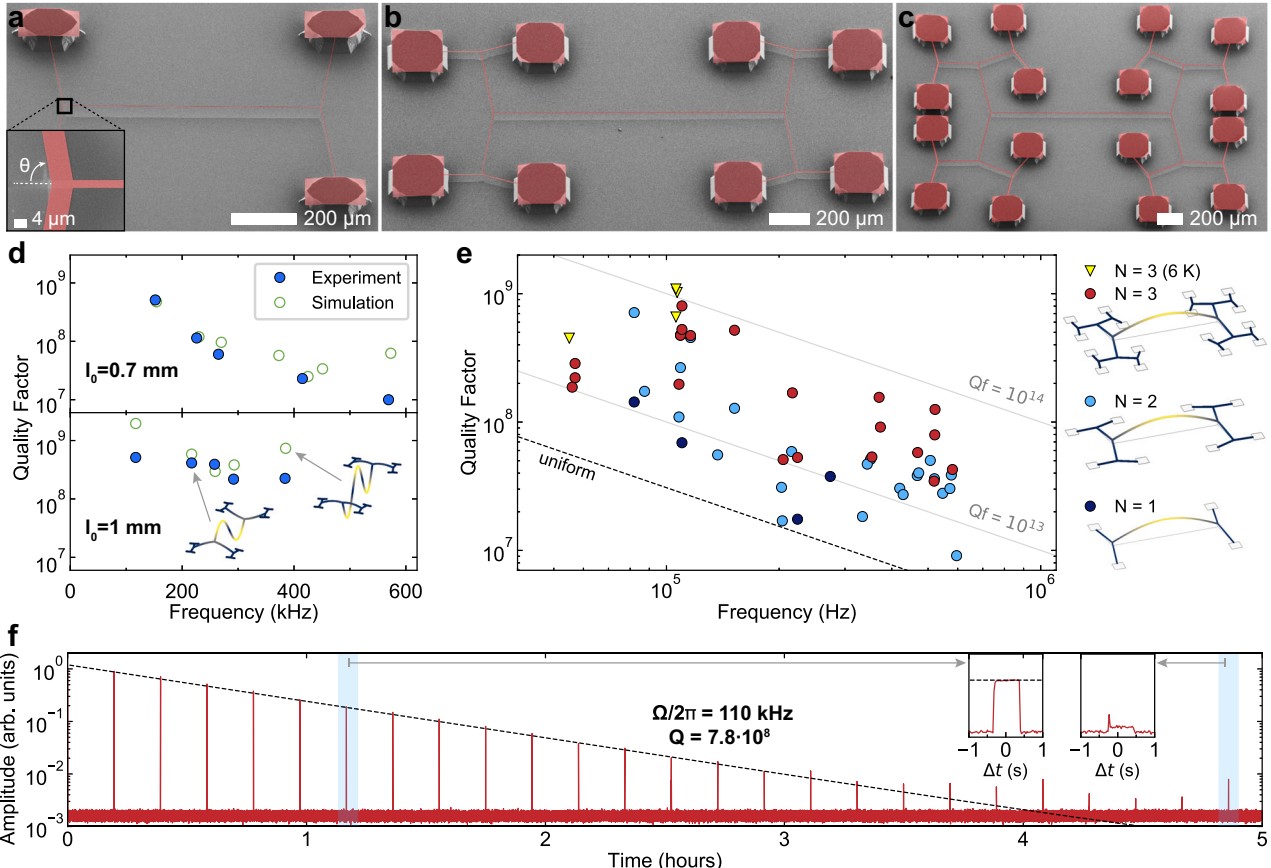

**Fig. 2 Binary tree nanobeam resonators. a–c** Scanning electron micrographs of binary tree beams with varying branching number, $N$. $Si_3N_4$ is shown in red, Si in gray. Inset of **a**: close-up of a bifurcating junction. **d** $Q$ versus mode order of two resonators with $N = 3$. Top: $\theta = 78°$, $r_l = 0.65$, and $l_0 = 0.7$ mm. Bottom: $\theta = 83°$, $r_l = 0.45$, and $l_0 = 1$ mm. Blue circles: measurement results. Green open circles: finite element model prediction. Insets show simulated mode shapes. **e** Survey of fundamental mode quality factors of binary tree beams. Colors differentiate $N$; circles indicate room temperature measurements, triangles correspond to measurements at about 6 K. The legend shows the evolution of the mode shape as a function of $N$. Dashed line: predicted fundamental mode $Q$ of a uniform beam with the experimentally observed intrinsic dissipation. **f** Sample ringdown trace (red) and exponential fit (black). The measurement was performed at room temperature. Inset: close-ups of two intervals when the measurement laser was on (~0.7 s-long).

with $N > 3$ and large branching angles, $\theta > 60°$, are challenging to fabricate, because of the segment width growth in higher branching generations, and due to spatial constraints of the pad supports (see Fig. 2c).

We could consistently observe devices with room-temperature $Q > 5 \times 10^8$. As expected, increasing the number of branching generations, $N$, increased the quality factor while leaving the fundamental mode frequency approximately constant. This trend was observed up to $N = 3$. The quality factors of low-order modes of $N = 3$ devices were generally in good agreement with theory, as shown by the data in Fig. 2d. This data also confirms the theoretical prediction that many low-order modes of binary tree beams experience reduction of dissipation by soft clamping at the same time. Some discrepancy between experiment and theory was especially evident at the low end of the explored frequency range. The devices with 57 kHz fundamental modes had quality factors more than ten times lower than the predicted values. This discrepancy could be explained by the influence of other loss mechanisms besides internal friction, such as acoustic wave radiation to the chip bulk and fabrication imperfections caused by wide branching segments. One potential extrinsic loss mechanism—damping by residual gas in the vacuum chamber—was experimentally ruled out at room temperature ($Q_{gas} \approx 10^{10}$ for a 20-nm-thick resonator at 57 kHz and pressure $= 5 \times 10^{-9}$ mbar)[29]. Further investigations are required to understand the origin of added losses.

Cryogenically cooling the binary tree resonators down to 6 K moderately increases their quality factors, up to values as high as $Q = 1.1 \times 10^9$. Under cryogenic conditions, the resonators also attain their highest thermal noise-limited force sensitivity, which reaches $\sqrt{S_F} = \sqrt{4k_B T m \Gamma_m} \approx 90$ zN/Hz$^{1/2}$ for the highest-$Q$ mode with a 38 pg effective mass.

The self-similarity found in binary tree resonators leads to distinctive spectral features. Acoustic mode densities are convenient to characterize using the cumulative distribution function,

$$\mathrm{CD}(\omega) = \sum_{n:\Omega_n \leq \omega} 1, \tag{2}$$

which gives the total number of modes below the frequency $\omega$. For a homogeneous medium in which the frequency of acoustic waves is proportional to their wavevector, CD is proportional to $\omega^d$, where $d$ is the spatial dimension. This scaling explains mode densities in simple resonators, such as tensioned strings (where $\mathrm{CD} \propto \omega$) and membranes (where $\mathrm{CD} \propto \omega^2$[30]). Structural self-similarity can break this rule while preserving the power law behavior $\mathrm{CD} \propto \omega^{\tilde{d}}$ for low-order modes. In this case, the power $\tilde{d}$ can be fractional, and is called the spectral dimension of the structure[8,31].

Our experimental investigation of mode densities is outlined in Fig. 3. Figure 3a shows a binary tree resonator with $N = 4$ generations of junctions, $\theta = 60°$, and $r_l = 0.6$. The branching angle of the device was chosen such that the constant stress condition is fulfilled when all segments have the same width (2 μm for this device). The resonator's fundamental mode has a frequency of $\Omega_m/2\pi = 82$ kHz and $Q = 2.8 \times 10^8$ at room temperature (in good agreement with the theoretical prediction of $\Omega_m/2\pi = 82.5$ kHz and $Q = 3.0 \times 10^8$). We computed the cumulative distribution function of this device by measuring thermomechanical noise spectra at different points (marked in Fig. 3a) and counting the modes. The result is presented in Fig. 3c. The experimental mode distribution follows a power law with $\tilde{d} = 1.63$, in good agreement with the simulated distribution for which $\tilde{d} = 1.65$. In the simulation, only out-of-plane modes were counted, as our optical interferometer is insensitive to in-plane modes. We evaluate our procedure by measuring the cumulative mode density of a uniform beam ($\tilde{d} = 1$) and a square membrane ($\tilde{d} = 2$) and find for each the expected power law scaling.

The excess of vibrational modes of binary tree resonators above those of a one-dimensional structure is mainly due to modes localized in high-generation segments as shown in Fig. 3b. We can confirm the localized nature of some low-order modes by comparing the thermal noise spectra taken at the 0th and the 1st generations of segments. The two spectra shown in Fig. 3d indicate that the mode around 230 kHz, marked by the rightmost dashed line, has no amplitude on the 0th generation segment, and hence is localized. At higher frequencies than those in Fig. 3d we cannot reliably establish a correspondence between the simulated and the measured spectra (see Fig. 3e), but, remarkably, the experimental mode distribution still follows the predicted power law. This deviation might be caused by mode coupling and stress non-uniformity due to fabrication imperfections.

While binary tree beams have outstandingly low dissipation and mass, two-dimensional membrane resonators exhibit a higher interaction efficiency when they are embedded in Fabry-Pérot cavities or interrogated interferometrically. The integration of a nanomechanical membrane in an external optical cavity has enabled a range of optomechanics experiments[20,27,28].

We apply hierarchical junctions to trampoline membranes, a type of resonator that was introduced recently[32,33], and inspired numerous applications[34–36]. We implement two designs: the first one consists of two orthogonal binary trees combined with a pad, as shown in Fig. 4a; the second design, which we term a 'steering wheel' membrane, can be produced from the first one by pairwise joining half of the segments at the 2nd branching generation (Fig. 4b). Steering wheels are particularly simple to fabricate, but have effectively only one generation of branchings, limiting the achievable dissipation dilution levels. Interestingly, similar geometries were obtained in ref. [37] through a topology optimization algorithm, and experimentally demonstrated in ref. [38]. Integration of the self-similar trampolines with a backside window is possible but complicates the fabrication process; for this reason, in the device displayed in Fig. 4a, the silicon substrate is still present below the trampoline. In the Supplementary Information we provide more details on the potential inclusion of a backside window. In both our membrane designs, the junctions are not exactly self-similar, and the width profiles of their segments are fine-tuned to optimize quality factors while maintaining constant stress. In the fabricated devices, the pad size varies between 35 and 85 μm.

The fundamental modes of our trampolines are partially soft-clamped, which can be seen from the suppression of mode gradients towards the peripheral clamping points in Fig. 4e, f. The experimental results for devices with lateral extent between 0.5 mm and 2 mm are summarized in Fig. 4g. We observed quality factors of fundamental trampoline modes as high as $Q = 2.3 \times 10^8$ at $\Omega_m/2\pi = 100$ kHz and $Q = 1.7 \times 10^7$ at $\Omega_m/2\pi = 470$ kHz. We remark that the self-similar trampoline membrane exhibited a $Q$ about 3.5 times lower than the finite element prediction (red symbols in Fig. 4g), potentially due to out-of-plane static deformations that we observed in the suspended device (more details are included in the "Methods" section).

These dissipation dilution levels are a factor of three beyond those of the original trampoline designs[32] at the same frequencies. Trampolines with branching tethers, with a smaller pad (about 35 μm) and lower effective mass (260 pg) than steering wheels, exhibit $\sqrt{S_F} \approx 3.7$ aN/Hz$^{1/2}$ at room temperature.

We experimentally demonstrated resonators which attain quality factors up to $7.8 \times 10^8$ through the combination of dissipation dilution with hierarchical design. The dissipation dilution is highest at low frequency, extending the soft clamping technique to the fundamental and low-order flexural modes.

The dissipation of binary tree beams at the lowest frequencies in the examined range is significantly higher than theory predictions.

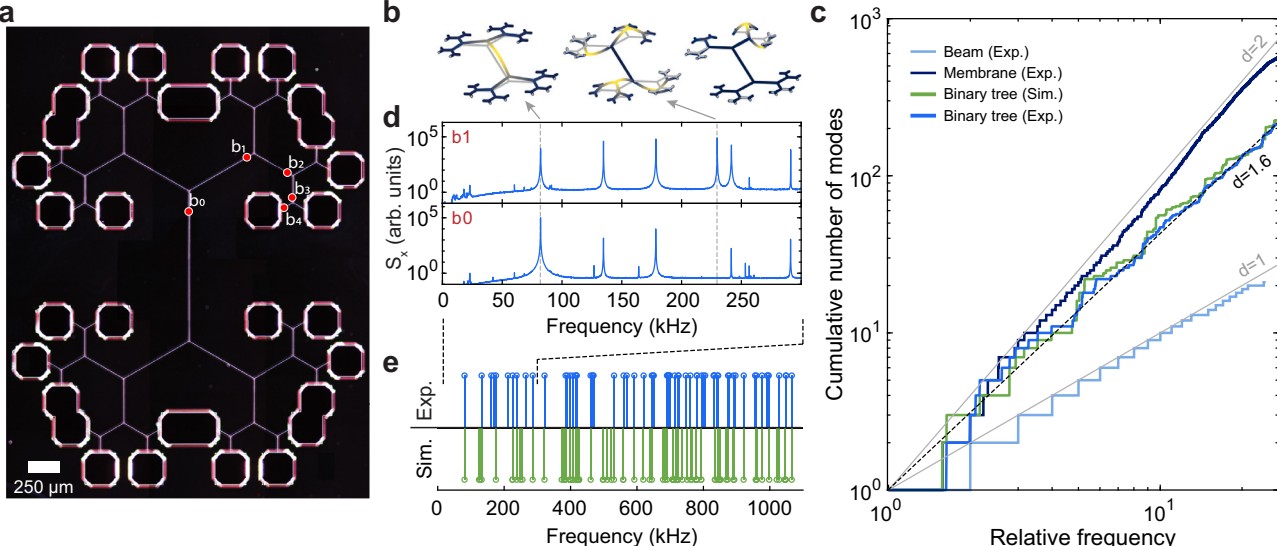

**Fig. 3 Mode density of self-similar mechanical resonators. a** Dark field optical image of the measured device. **b** Simulated mode shapes of the mechanical modes indicated in the thermomechanical spectra of **d**. **c** Cumulative mode density measured for a uniform beam, a binary tree beam and a square membrane. For the binary tree beam, we also show the cumulative mode density expected from simulation. The gray lines indicate linear and quadratic scaling. **d** Thermomechanical noise spectra acquired in low vacuum by optical readout at the points indicated in **a**. **e** Comparison of simulated and measured resonant frequencies. The indicated interval corresponds to the frequency range in **d**.

If additional loss mechanisms are identified and suppressed, our highest aspect ratio resonators could exhibit $Q > 10^9$ at room temperature. Going further, even higher dilution levels could be achieved by introducing strain engineering[15] in the hierarchical structures. Higher tensile strain at the mode antinode could be induced by appropriately choosing the segment width ratios. The hierarchical principle may also be applied to enhance the dissipation dilution of tensioned pendula[10,39], enabling lower thermal noise for optical element suspensions used in gravitational wave detectors.

The thermal noise-limited force sensitivities of the best hierarchical resonators surpass those of state-of-the-art atomic force microscopy cantilevers[40], which makes them excellent candidates for use in 'inverted microscope' configurations in scanning force microscopy[25]. The high stress in our structures allows driving the oscillator to large amplitudes before the onset of anharmonicity[11,41], and may allow driven oscillations with very low frequency noise, potentially useful for mass sensing and imaging[42].

## Methods

**Stress and intrinsic loss in silicon nitride films.** The elastic and anelastic parameters of the employed $Si_3N_4$ films (from Hahn-Schickard Gesellschaft für angewandte Forschung e.V.) were estimated by characterization of an array of uniform beams with variable length, fabricated with the same process and machinery used for all the resonators in this work. Frequencies and quality factors of fundamental modes of doubly-clamped beams with uniform widths were measured and are displayed as a function of length in Fig. 5.

The deposition stress of the $Si_3N_4$ thin film, $\sigma_{dep}$, was determined by fitting the lowest-order flexural resonant frequencies with a linear dispersion model $\Omega_0/2\pi = \frac{1}{2l}\sqrt{\frac{\sigma_{dep}(1-\nu)}{\rho}}$, where $\rho = 3100$ kg/m$^3$ is the volume density of $Si_3N_4$, $\nu = 0.23$ its Poisson's ratio, $l$ the beam length and $\sigma_{dep}$ is the parameter to be determined. The fit result is $\sigma_{dep} \approx 1.03$ GPa. This value reliably predicted all the experimentally observed resonance frequencies of the beam and membrane-type resonators presented in our work.

The approximate expression for the resonant frequency is appropriate for the limit of negligible bending stiffness and is valid for high aspect ratio beams with $\lambda = \frac{h}{l}\sqrt{\frac{E}{12\sigma_{dep}(1-\nu)}} \ll 1$[11] ($h$ is the beam thickness and $E$ the Young's modulus of $Si_3N_4$). In the same limit, the dissipation-diluted quality factor will linearly increase as a function of string length, $Q = Q_{int}/2\lambda$. This trend is indeed found for beams with $l < 1.5$ mm in Fig. 5b: A fit to the $Q$ of these resonances provides an estimate of the $Si_3N_4$ intrinsic mechanical loss of $Q_{int} \approx 2500$. This value is two times higher than the average value of $Q_{int} = (1200 \pm 800)$ found in ref. [43] for surface losses in

$Si_3N_4$ (at the film thickness of $h = 20$ nm used for this work; in the surface dissipation-dominated regime $Q_{int}$ increases linearly with $h$). Longer beams deviate from the predicted linear trend of dissipation dilution, exhibiting a nearly constant $Q$ independent of frequency and length due to an unknown loss mechanism.

### Fabrication details

*Binary-tree beam resonators.* The fabrication process (see Fig. 6b) is based on our previous work on strain-engineered nanobeam resonators[15,44]. The first step in the process flow is low pressure chemical vapor deposition (LPCVD) of a 20 nm thin film of stoichiometric $Si_3N_4$ onto a silicon wafer. (100)-oriented wafers are needed to facilitate the undercut of $Si_3N_4$ while preventing undercut of the beam supports (see description below for details). The resonator geometry is defined in a first step of electron beam (e-beam) lithography (panel 1) using flowable oxide resist (FOX®16). The mask is transferred to the underlying $Si_3N_4$ by plasma etching using fluorine chemistry. The second step of e-beam lithography uses an up-scaled mask to ensure that the $Si_3N_4$ layer is fully encapsulated by the second FOX®16 layer. The encapsulation protects the $Si_3N_4$ layer in the subsequent deep reactive ion etch (DRIE) which creates a recess deeper than 15 μm (2). At this point the wafers are diced into chips sized 5 × 12 mm, with the devices protected from both damage and contamination by a 15-μm-thick layer of viscous photoresist. The chips are then cleaned in a piranha solution and the photo- and e-beam resists are stripped in baths of NMP (1-methyl-2-pyrrolidon) and buffered hydroflouric acid (BHF), respectively. In the final step, the $Si_3N_4$ chips are placed in a potassium hydroxide (KOH) bath at ~55 °C, which suspends the beams by removing the silicon below the resonators (3). The anisotropy of the KOH etch is exploited to ensure that the beams are fully released while the supports of the resonators remain largely intact, as the (111) planes are etched much more slowly than the (100) and (110) planes. The recess created in the DRIE step facilitates the undercut by exposing the fast-etching planes in silicon and thereby reducing the etch time required for releasing the beams. It is necessary to dry the released beams in a critical point dryer (CPD), since the resonators are extremely fragile due to their high aspect ratios. When dried mechanically, the resonators are likely to collapse due to strong surface tension at the liquid-air interface.

Limiting the undercut of the supports is a crucial design consideration for fractal-like beam resonators (see Fig. 6a) since any overhang of the supports was observed to deteriorate the quality factors. The branching angles of binary trees differ from multiples of 90°; therefore some beam segments will not be aligned along the slow-etching planes of silicon (viz. oriented along ⟨110⟩ directions). If the final beam segments are orthogonally clamped to their support pillars, the latter will be attacked from all directions during the KOH etch (see Fig. 6a). The removal of silicon below the supports and in the vicinity of the clamping points leaves this part of the $Si_3N_4$ film free to deform. We attribute increased dissipation to this deformation and the corresponding change in strain of the $Si_3N_4$ film, which is supported by our measurements: binary tree resonators with significant overhang at their connections to the frame were found to have quality factors more than one order of magnitude below the values predicted by our analytical model. Fortunately, it is possible to largely eliminate the overhang at the clamping points by aligning the

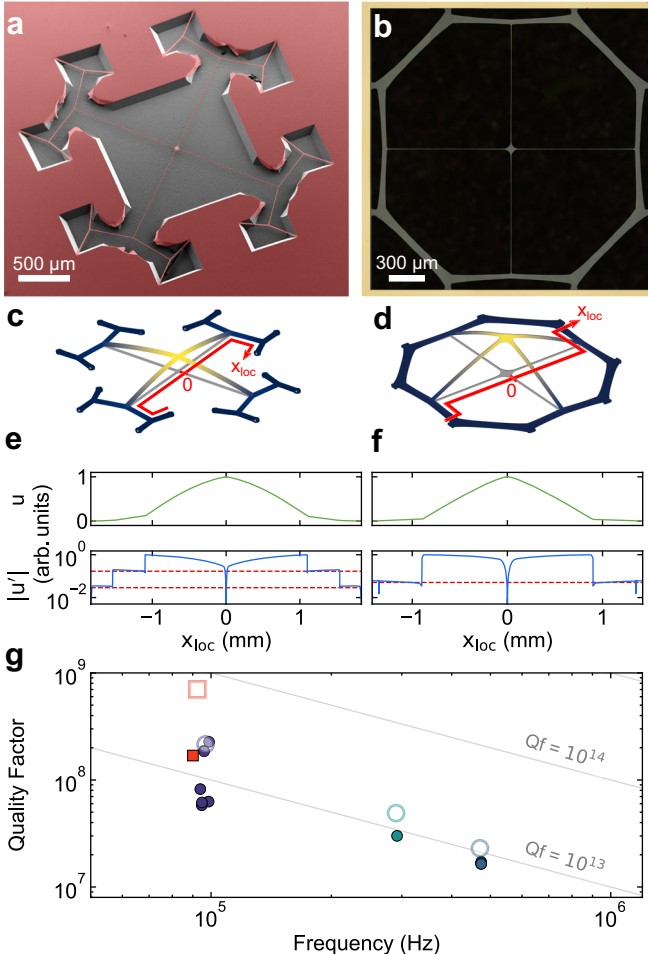

**Fig. 4 Trampoline membranes with partial soft-clamping. a** Scanning electron micrograph of a trampoline resonator with branching tethers. The silicon substrate is recessed by about 80 μm below the trampoline. **b** Optical microscope image of a steering wheel membrane, enclosed in a square frame of about 2.2 mm side length. The silicon substrate is completely removed below the sample. **c, d** Simulated displacement profiles of the trampoline and steering wheel fundamental modes. **e, f** Displacement profile (green) and its first derivative (blue) evaluated along the red paths highlighted in **c, d**. The red dashed line indicates the step-like gradient suppression after a bifurcation, by a factor of $\cos(\theta)$. **g** Measured quality factors of self-similar and steering wheel trampolines. Squares represent trampolines with branching tethers, while circles corresponds to steering wheels. Colors differentiate distinct designs. Measurements of individual devices are displayed with filled symbols, while open symbols portray the $Q$ factors predicted with a finite element model.

supports along the slow-etching planes (see Fig. 6a). In this case the silicon at the clamping points is protected during the KOH etch since only the slow-etching planes are exposed. The results reported in the main text were obtained for binary tree resonators based on this approach. It should be noted that the rotation of the supports inevitably leads to non-orthogonal clamping of the final beam segments causing a deviation from the ideal clamping conditions assumed in our model. This boundary geometry induces limited changes to the frequencies and dissipation rates of the lower order modes due to their limited mode amplitudes at the clamps, as confirmed by finite-element simulations. The increase in dissipation is small in comparison with the deleterious effect of a significant overhang at the supports.

*Trampoline membrane resonators.* Trampoline membranes with branching tethers are fabricated similarly to binary-tree beam resonators. The overall process follows the one described in the Supplementary Material of ref. [18] and in ref. [45]; here, we highlight the main differences with respect to the fabrication process of binary-tree beams. After patterning and etching the outline of the membrane resonators on the front side $Si_3N_4$ layer with e-beam lithography, we exploit the $Si_3N_4$ layer

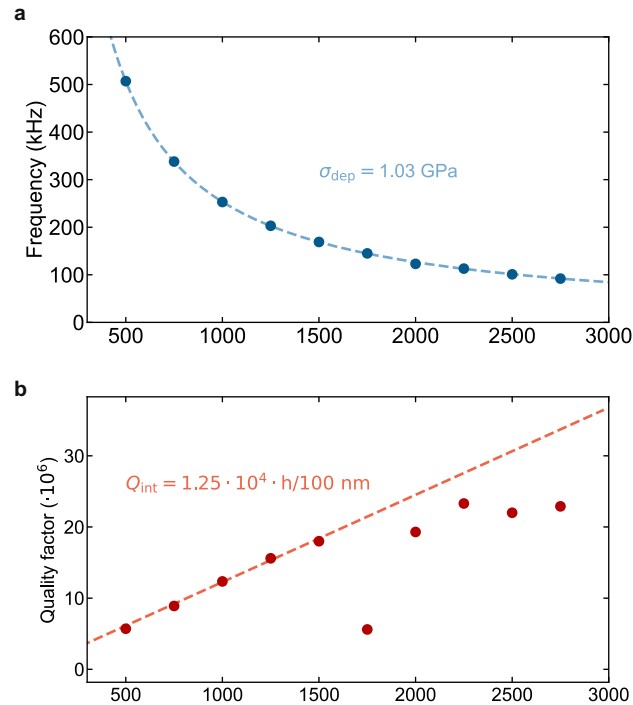

**Fig. 5 Determination of mechanical parameters of $Si_3N_4$ thin films. a** Frequency and **b** quality factors of uniform $Si_3N_4$ beams of variable lengths.

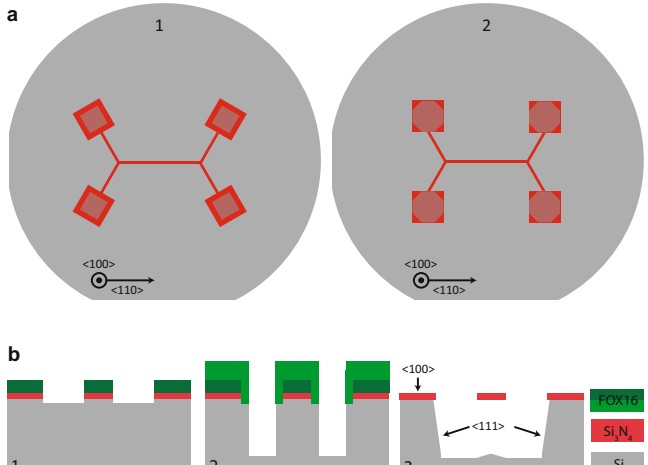

**Fig. 6 Design considerations in KOH-based undercut of $Si_3N_4$ on Si. a** Appropriate design limits the undercut of the beam supports and prevent deformation and relaxation of the $Si_3N_4$ film. The KOH etch fully removes the silicon below the beam (dark red), while the $Si_3N_4$ at the supports is still partly connected to the underlying silicon (light red). If the final beam segments are connected to their supports at a 90° angle (1), the supports will be undercut at the clamping points, creating an overhang region which increases dissipation (see main text). Aligning the supports along the slow-etching planes (2) ensures that the undercut is limited to the corners of the pads and prevents the creation of an overhang at the clamping points. **b** Main steps of the process flow.

concomitantly deposited on the back side as a mask for the KOH undercut of the devices, and pattern rectangular membrane windows with UV lithography, aligned to the front side through appropriate reference markers. During this step, a layer of photoresist protects the front side layer from contamination and mechanical damage by contact to the wafer holders of the optical exposure and plasma etching tools. The dimensions of the windows are corrected with respect to the membrane

outline on the front side to take into account the shrinking of the window as the etch progresses through the wafer thickness, owing to the slow-etching (111) planes, non-orthogonal to the wafer surface, being uncovered. Precise knowledge of the wafer thickness and optimal alignment to the front side patterns are required to avoid the formation of substantial overhang at the tether clamping points.

The KOH etch is typically subdivided in two steps. The first step removes the bulk of the silicon thickness (≈10–40 μm of silicon is retained) and occurs from the back side only, while the front side layer is protected by an appropriate sealing wafer holder[18]. The second step is performed after chip separation, exposing both sides of the chips to the KOH etch. This bulk etch process removes the need for the DRIE step performed on beam samples. According to the specific tether clamping design, the two KOH steps must be timed carefully, in order to fully release the tethers without creating overhang at the clamped edges.

Membrane chips are dried from their rinsing baths though CPD as well: despite the low impact of stiction forces by virtue of the presence of an optical window through the whole wafer thickness, we found CPD useful to increase the device yield and decrease the residual contamination associated with the wet etch steps.

**Finite elements simulations.** This section is adapted from ref. [46] and is included here for completeness. We simulate the dilution factors $D_Q$ and effective masses $m_{eff}$ of the resonator modes using finite element methods (FEM). We perform pre-stressed eigenfrequency analyses in COMSOL Multiphysics, conducting 2D simulations with the "Shell" interface.

We mesh the branch domains with anisotropic rectangular elements and refine the mesh in the vicinity of the clamping points, in order to capture accurately the clamping curvature[13]. We enforce fixed boundary conditions at the clamping points, i.e., $u = \partial u / \partial \vec{n} = 0$ (where $u$ is the displacement field and $\partial u / \partial \vec{n}$ its derivative normal to the boundary).

We evaluate the effective mass with respect to a point-like laser probe centered on the maximum of the displacement field:

$$m_{eff} = \frac{\rho h \int u^2\, dx dy}{\max(u^2)}, \qquad (3)$$

where $\rho = 3100$ kg/m$^3$ is the density of Si$_3$N$_4$. The dilution factors are calculated with the ratio of kinetic and linear elastic energies[11]. For out-of-plane and torsional resonances:

$$D_Q = \frac{12\rho\Omega_m^2}{Eh^2}(1-\nu^2)\int w^2\, dx\, dy \, / $$
$$\int \left(\partial^2 w/\partial x^2 + \partial^2 w/\partial y^2\right)^2 \qquad (4)$$
$$+ 2(1-\nu)\left(\left(\partial^2 w/\partial x \partial y\right)^2 - \partial^2 w/\partial x^2 \times \partial^2 w/\partial y^2\right)dx dy,$$

where $w$ is the $u$ component in the out-of-plane direction. We then use the measured $Q_{int}$ to infer the simulated $Q$ factor as $Q = D_Q \times Q_{int}$.

**Buckling and static deformations in suspended structures.** Structures with internal or external compressive loads are subject to buckling, an elastic instability phenomenon that occurs when local stress surpasses a threshold value[47]. In thin Si$_3$N$_4$ structures (in our work, $h \approx 20$ nm), buckling is commonly observed as a static out-of-plane deformation, which releases elastic energy. The dissipation dilution in buckled resonators is severely degraded.

While no significant buckling was observed in binary tree beam resonators, it was frequently encountered in trampoline membranes with non-optimized designs. In Fig. 7a, an example static deformation pattern is portrayed, which occurred in a steering wheel membrane close to its clamping point. A FEM simulation of the static stress distribution in the same structure is presented in Fig. 7b, which shows that the buckling is co-localized with regions of compressive principal stress. To obtain the color map in the figure, we decompose the simulated in-plane stress tensor into its principal components (i.e., its eigenvalues) and display the smallest one. Wherever one principal stress is negative, there is compression along some direction, which can trigger buckling. In the region where buckling is observed in the real device, the simulated minimum principal stress dips to $\approx -30$ MPa.

Our strategies to avoid buckling in trampoline membranes were (a) to ensure that the structures are stress-preserving[7], i.e., that the forces acting on each junction point are balanced prior to suspension, and (b) to taper wide segments, which helped maintain their transverse stress above zero. Buckling is also expected to be less significant for films thicker than 20 nm.

Another type of static deformation that was observed in our tensile resonators is twisting of film segments. An example is portrayed in Fig. 7c for the outer-most generation of segments of a trampoline membrane with branching tethers. The sample topography has been obtained through a confocal microscope profilometer. We conjecture that potential causes for this phenomenon could be the inhomogeneity of the film stress in the vertical direction, or the non-uniformity of the height of the silicon wafer. We expect that these undesired static deformations would be reduced by increasing the film thickness.

**Interferometric mechanical characterization setup.** All the measurements presented were performed using a balanced Mach-Zehnder homodyne interferometer

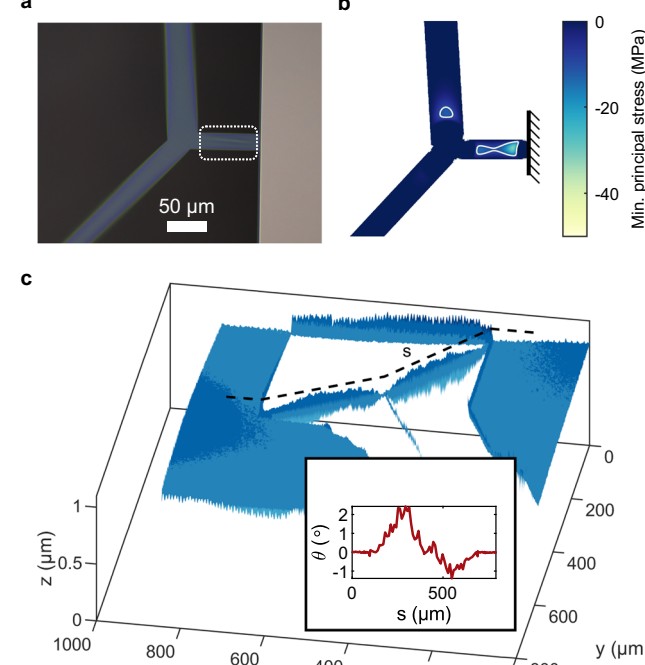

**Fig. 7 Buckling in a steering wheel resonator. a** The branch connecting the membrane to the silicon frame is visibly wrinkled (in the region circled with a dashed line), while wider tethers are bent in the transverse direction, as can be seen by the tether shifting out of the focal plane of the microscope. **b** Minimal principal stress component from a FEM simulation of the membrane in **a**. The area where buckling is most prominent exhibits large compressive stress in the direction transverse to the branch. White contours encircle the regions with compressive stress below −10 MPa. **c** Profilometry of a trampoline membrane sample exhibiting static torsion in the branches closest to the clamping points. Inset: rotation of the beam surface normal along the red dashed path. The silicon frame of this sample shows a height variation up to 1 μm/mm.

(see Fig. 8). In the signal arm of the interferometer, the light beam is focused through a microscope objective or a simple convergent lens on the sample under examination, and a small fraction (0.1–10%) of the impinging optical power is collected in reflection and steered on a beam splitter where it interferes with a stronger local oscillator beam ($P \approx 1$ to 10 mW).

The samples and a 3-axis nanopositioning stage, employed for alignment, are housed in a vacuum chamber capable of reaching pressures below $10^{-8}$ mbar, sufficient to eliminate the effect of gas damping for the devices measured in this work. A piezoelectric plate, used to resonantly actuate the devices, is connected to the sample mount.

The low frequency fluctuations of the interferometer path length difference are stabilized by means of two cascaded acousto-optic modulators (AOMs): the first shifts the frequency of the local oscillator beam by +100 MHz, while the second shifts it back by −100 MHz, with a small frequency offset controlled using the low-frequency signal from the balanced photodetector couple as an error signal. With this arrangement, the phase difference can be tuned finely with practically unlimited actuation range. The phase difference is stabilized close to phase quadrature, corresponding to the largest transduction of mechanical displacement.

Ringdown measurements are initiated by exciting a mechanical resonance with the piezoelectric plate, then turning the mechanical excitation off abruptly and recording the slowly-decaying amplitude of the displacement signal, obtained by demodulating the photocurrent signal at the excitation frequency. A demodulation bandwidth exceeding 100 Hz is employed, to mitigate the effect of mechanical frequency drifts, induced e.g., by temperature or optical power fluctuations. As detailed in the following, gated measurements are performed in order to minimize the influence of optical backaction of the probing beam; to this end, a mechanical shutter inserted at the output of the laser, is actuated periodically.

**Acquisition and analysis of binary tree resonators mode density data.** The spectral properties of the structure are found by acquiring thermomechanical spectra via optical interferometry and identifying the mechanical modes with a peak-finding algorithm. The cumulative distribution function is then computed by

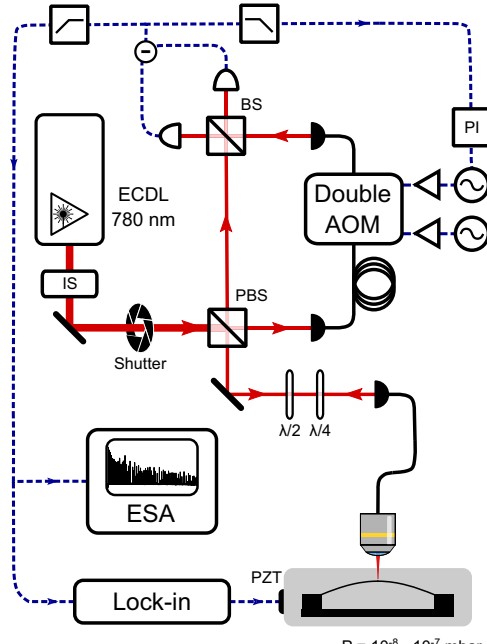

**Fig. 8 Simplified scheme of the mechanical characterization setup.** ECDL external cavity diode laser, IS intensity stabilizer, PBS polarizing beam splitter, BS 50–50 beam splitter, AOM acousto-optic modulator, $\lambda/2$ and $\lambda/4$ half-wave and quarter-wave plate, PI proportional-integral feedback controller, PZT piezoelectric actuator, ESA electrical spectrum analyzer.

counting peaks below a certain frequency. Spectra are acquired at two points on all the branches on one half of the tree (62 spectra in total, including the fundamental branch), as the interferometric thermomechanical signal depends on the mode amplitude at the focus location of the probe beam. As optical driving or damping of the mode is not a particular concern for this measurement, it is performed using higher probe power ($\approx 5$ mW) than the ringdown measurements. However, due to the narrow linewidth of the fundamental mode, high optical power frequently results in its excitation and appearance of its higher harmonics in the spectra. To avoid this problem, the pressure in the vacuum chamber was increased to $10^{-7}$ mbar to damp the fundamental mode.

Data analysis is then performed as follows: first, Lorentzian peaks with a minimum prominence and a minimum frequency spacing are identified on each spectrum. Then we take the union of all the sets of frequency values from all the branches. To avoid double-counting of the modes, the union is done within a frequency tolerance, meaning that peaks closer than a fixed frequency separation are counted as one, as jitter or thermal drifts cause small changes in the resonant frequencies among different spectra. We also take several spectra from the substrate and $Si_3N_4$ pad surfaces and exclude their peaks from the unified set of mode frequencies. This final 'purified' set is counted and shown in Fig. 3e, where it is compared to flexural out-of-plane modes obtained from the FEM simulation. The same procedure is used to acquire the mode densities of the uniform beam and the square membrane.

## Data availability

The data generated in this study and supporting the manuscript figures (ringdown measurements and thermomechanical spectra), as well as the lithographic fabrication masks have been deposited in the following open-access Zenodo database: https://doi.org/10.5281/zenodo.5873960.

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

## Acknowledgements

The authors thank Matthieu Wyart for scientific discussion and Guanhao Huang for experimental assistance. This work was supported by funding from the Swiss National Science Foundation under grant agreement no. 182103, the EU H2020 research and innovation programme under grant agreement no.732894 (HOT) and the European Research Council grant no. 835329 (ExCOM-cCEO). N.J.E. acknowledges support from the Swiss National Science Foundation under grant no. 185870 (Ambizione). This work was further supported by the Defense Advanced Research Projects Agency (DARPA), Defense Sciences Office (DSO) contract no. HR00111810003. All samples were fabricated at the Center of MicroNanoTechnology (CMi) at EPFL.

## Author contributions

M.J.B. and R.G. fabricated the binary tree resonators with support from S.A.F., N.J.E and A.B.; A.B. fabricated the membrane resonators with support from S.F.; S.F. and A.B. developed the membrane designs with support from N.J.E.; the devices were characterized by A.B., R.G., N.J.E., M.J.B. and S.F.; A.A. acquired and analyzed the mode density data with support from N.J.E and S.F.; the manuscript was written by A.B., S.F. and N.J.E. with support from the other authors; the work was supervised by N.J.E and T.J.K.

## Competing interests

The authors declare no competing interests.
