## [Peer review file · Nature Communications]

REVIEWER COMMENTS

Reviewer #1 (Remarks to the Author):

The authors present an experimental work that follows their computational investigation published in reference [9]. I agree with the assessment from previous reviewers, highlighting that this article has less impact because of the existence of reference [9]. However, in my opinion the fabrication and testing of this concept is still an interesting finding and the quality factor that was achieved at a low frequency mode is impressive.

The key question is whether this is enough to recommend it for publication in Nature Communications or not. In my opinion it is enough, once the authors consider the following points:

1. The authors should provide a quantitative evaluation of the loss around the joints, so that their impact on the performance of the resonator is clarified. My understanding is that the rapid curvature change concentrated on the clamping points for the straight beam is spread out to the joints of the hierarchical structure to reduce dissipation. Considering Fig. 1c, the sharp displacement change close to the joints introduced new types of damping at the hinges, reducing the clamping effect significantly, i.e. using multiple joints to alleviate the sharp curvature changes.
2. Why does the Q drop for the design in Fig. 4 when compared with Fig. 2? Could it be the effect of the center pad?
3. The authors claim they have matched the stress for all beams (constant tension throughout the structure to minimize static in-plane deformations upon structure undercut). Does this help maximizing Q? If not, I recommend to clearly state that this is only related to fabrication.
4. In Fig. 2d the authors mentioned that gas damping could be the dominant source for the difference between experiment and simulation. I recommend to also mention that gas damping is higher at a lower frequency. In addition, the conclusion section mentions that it would be possible to exceed a billion Q in room temperature if suppressing other loss mechanisms. It would be relevant to explain how this can be achieved, for example can gas damping be reasonably decreased with improved vacuum?
5. Also in Fig. 2d, the 0.7 mm resonator at high frequency exhibits a strong mismatch between simulation and experiment. What is the reason for this?

6. I recommend to add more details about the finite element model, such as the mesh size and the simulation time, among others. Even if some of these details can be found in reference [9], at least be explicit about it.

7. In the methods section, the intrinsic Q value is mentioned to be two times higher than the average value present in other literature. Why is this the case?

8. It might be relevant to include an additional reference (I am not an author of this reference, neither do I have any connection to the authors) that performed the experimental validation of the topology optimization paper [ref. 39 in the article under review] and that was also published in Nature communications 12, 5766 (2021).

Reviewer #2 (Remarks to the Author):

In this work, the authors experimentally verify their nanomechanical designs from a previous publication [ref. 9] with established fabrication and ring-down measurements. In doing so, the authors attain Qs of 780 million with a fundamental mode (at 300K) and > 1 billion (at 6k). The paper is thorough in terms of its characterization and theory and expands the use of their designs to membrane structures, which is novel with respect to ideas presented in ref. 9. What is dubious is how this work fits in within the existing state-of-the-art literature and the impact it can have on the community given that there are alternatives in the field that operate in the same regime with similar (if not better) performance. It is also not so clear how well these devices can be realistically implemented into experiments. As Nature Communications aims to publish state-of-the-art research which can make an impact on the field, it would be important for the authors to address the following points before considering for publication:

1. Higher f and Q are ultimately preferred for quantum optomechanics experiments – one the authors' motivations for the hierarchical resonators. If one requires higher frequencies (~ 1 MHz) and higher Q , one can look towards nanostring resonators published by the same group (ref. 25, Gadhimi et al.) where the frequency is nearly 10x higher and Qs larger by 100 million than ones shown in this study. The main argument the authors make for a hierarchical resonator (over resonators with phononic bandgap structures) is that hierarchical resonators uses a fundamental mode which has a large frequency spacing between it and next adjacent mode. However, the higher order modes in nanostrings (like those in ref. 25) can attain a similar relative frequency spacing between the mode of interest and the next adjacent mode via the phononic bandgap. While lower order modes conventionally equate to larger frequency spacing, phononic bandgap structures evade that rule by creating a large bandgaps around higher-order modes. It is not so much the mode number that is important, but the spectral spacing which is achievable between the mode of interest and adjacent modes. In this sense ref. 9 does this just as well –

but with a higher Q and frequency which is an order of magnitude larger! It seems phononic bandgap resonators are still significantly more desirable (than hierarchical structures) for quantum optomechanics experiments. Can the authors comment on this?

2. If one wants similar “low” frequency resonators (on the order of $\sim 100\text{kHz}$) and higher Qs, the same group of this study have recently published ‘perimeter mode’ nanomechanical resonators, arxiv:2108.03615 (2021) with $Q > 3$ billion at similar frequencies, and another group (Advanced Materials (2021): 2106248) has also demonstrated structures $\sim 100\text{kHz}$ and $Q > 10^9$, both in room temperature. These ‘perimeter mode’ resonators also have similar sizes as hierarchical resonators and higher Qs. Although they do not use fundamental modes, the mode spacing are comparable which is the big reason to target the fundamental mode. Given the similar operating regime it would make sense to discuss in terms of how hierarchical structures fit into the existing state-of-the-art.

3. One of the prominent differences between this study and designs in ref.9 is the use of hierarchical structures to produce trampoline membranes. The main motivation for membrane structures is to interface the central pad with large optical beam paths or optical cavities (in a way that nanostring cannot). Here it should be noted that the steering wheel structure shown in Fig 4b is very similar to topologically optimized resonators which were recently demonstrated in Nat. Communications (12, 5766 (2021)). Can the authors comment on any fundamental differences or similarities?

4. In Fig 4a, the authors show a “self-similar” trampoline with Q that is 60% larger than that shown in the Nat Comms 12, 5766 (2021) and the steering wheel trampoline, but it is only etched $80\mu\text{m}$ into the silicon substrate. Without etching through the substrate, it would not be possible to optically access the pad or interface with the resonator with an optical cavity (i.e. the main purpose of a pad). Is there a reason this was not etched all the way through the substrate? Is the fabrication prohibitively difficult? At the moment it is not discussed at all and would be important for researchers considering to implement themselves. I would imagine the the edges of self-similar patterning would not line up well with the square windows etched into the silicon by KOH. Would the extra overhang somehow affect the performance? It is important to note these limitations in fabrication since it would indicate whether the resonators are realistic and viable for researchers wanting to implement such structures into optomechanics experiments. As shown, it does not seem like it is currently possible to interface ‘self-similar’ trampolines into high-finesse cavities, making their usefulness limited for such applications.

5. One of the promises of hierarchical structures from ref. 9, is that with higher number of branchings, N, one can attain higher Qs. In ref. 9, $N = 4, 5$ are investigated and show higher Qs. However, in Fig 2, the author’s experimental progress stops at $N = 3$ without much explanation. This would be very important to comment on since it touches on the a possible limitation of this design in real-life implementation. Adding a discussion related to this would be very interesting for readers.

6. For clarity, I highly recommend matching the numbers describing the quality factor in the abstract. The experimental results in this paper (which is the essence rather than the simulation) show 0.78 billion for the room temperature Q and 1.1 billion for the cryogenic temperature Q. Stating the value more precisely will help the paper be more straightforward for readers. (ex. 7.8×10^8 Q for room temperature, 1.1×10^9 for cryogenic). Mixing in simulated Qs, with experimental numbers is very confusing for readers.

To conclude: On its own, the authors' paper presents concepts which are very interesting and well organized. Looking at the broader literature, my impression is that there are existing systems which could operate with similar (and sometimes better) performance at high Q & f, or, low f & high Q and large mode spacing. The hierarchical designs are definitely creative but not new given ref. 9 -- this leads one to assume that the advance must be in the fabrication or characterization techniques but these seem identical to ref. 25 (from the same group). If there is an advance here, the authors have not highlighted this. The question is whether putting them together signifies a significant leap in research. I would say yes if the devices represent state-of-the-art that is not achievable by other systems. This is the highest Q achieved for a fundamental mode, but the mode spacing achieved by this is comparable to other systems which do not use a fundamental mode. As presented it is not clear what is the significant leap that would merit publication in Nat. Comms or whether hierarchical resonators fill a gap that is not possible with existing systems. Addressing the points above could help to clarify this and improve the paper's significance.

Reviewer #3 (Remarks to the Author):

The authors have addressed the concerns expressed by the reviewers in a fully satisfactory manner.

I only noted the following detail: in their reply letter, the authors suggest to change the wording in the conclusion to "The high stress in our structures ..." whereas in the revised manuscript it still reads "The high dissipation dilution in our structures ...". This should be fixed.

Then I think this will be an excellent contribution to Nature Comms.

Reviewer 1:

The authors present an experimental work that follows their computational investigation published in reference [9]. I agree with the assessment from previous reviewers, highlighting that this article has less impact because of the existence of reference [9]. However, in my opinion the fabrication and testing of this concept is still an interesting finding and the quality factor that was achieved at a low frequency mode is impressive.

The key question is whether this is enough to recommend it for publication in Nature Communications or not. In my opinion it is enough, once the authors consider the following points:

We would like to thank the reviewer for their consideration and detailed evaluation of our manuscript. In the following sections we address the comments and questions raised by the reviewer (blue sections) and describe the amendments to the manuscript (red sections).

1. The authors should provide a quantitative evaluation of the loss around the joints, so that their impact on the performance of the resonator is clarified. My understanding is that the rapid curvature change concentrated on the clamping points for the straight beam is spread out to the joints of the hierarchical structure to reduce dissipation. Considering Fig. 1c, the sharp displacement change close to the joints introduced new types of damping at the hinges, reducing the clamping effect significantly, i.e. using multiple joints to alleviate the sharp curvature changes.

We thank the referee for this comment. The apparent discontinuity in displacement curvature at the joint is an artifact of the representation method we chose, plotting the displacement along a broken line that follows the hierarchy of branches (see Fig. 1c). The 3D displacement profile is actually smooth (see also the detailed discussion in [1]). Correspondingly, no significant increase in bending loss occurs at the joint: the residual losses of our hierarchical structures are well described by the combination of residual curvature at the clamping-points and lossy torsional motion of the branches (as explained in [1]). We reproduce below, for convenience, Fig. 3 of Ref. [1], that clarifies quantitatively that the losses due to distributed bending (β_{Bend}) are negligible compared to torsional losses (β_{Tors}), for the binary tree beams presented in this work with $\theta > \pi/3$. The dependent variables in the figure below refer to the following notation for the dissipation dilution factor: $D_Q = \frac{1}{\alpha\lambda + (\beta_{\text{Bend}} + \beta_{\text{Tors}})\lambda^2}$, with $\lambda \approx 10^{-5} - 10^{-3}$ for the binary tree beam devices.

Amendments to the manuscript

We added the following clarification regarding the mode displacement profile to the manuscript: “The abrupt mode gradient changes appear due to plotting the mode along a broken line that follows the hierarchy of the branches (red arrow path in the 3D model). As illustrated in the 3D mode profile the displacement profile undergoes a smooth transition towards the clamping points”

2. Why does the Q drop for the design in Fig. 4 when compared with Fig. 2? Could it be the effect of the center pad?

There are two membrane designs in Fig. 4: steering wheels and self-similar trampolines. The main reason for the reduced Q in the steering wheel design compared to binary trees is the adapted design to facilitate the fabrication of the trampolines. It can be seen in Fig. 4b, that the steering wheels are only limited to two branches and the branches are adjusted (joined together, breaking the self-similar condition) such that they maintain the same stress and fit in a rectangular frame to facilitate the backside etching process in the Si substrate; as a result, the clamping-points curvature is only partially suppressed compared to the distributed curvature. We mention in the text that these resonators are “partially soft-clamped” to emphasize on this point.

Self-similar membranes (Fig. 4a) do not experience this limitation and could in theory perform similar to the binary tree nanobeams (the numerically-predicted Q for the device reported in the manuscript is ~ 800 million, as display by the red square Fig. 4g). However, the main drawback of this design is the challenging fabrication, which resulted in a low yield of suspended devices. The structure in Fig. 4a was suspended from the wafer frontside to simplify undercut and SEM imaging; opening of a backside is possible and was demonstrated for other trampoline devices (see SEM image included in the reply to Reviewer 2), but makes the fabrication slightly more involved.

Amendments to the manuscript

We added more clarification regarding lower Q factors of the steering wheel membranes compared to the binary tree beams (“Steering wheels are particularly simple to fabricate, but have only one generation of branchings, limiting the achievable dissipation dilution levels”) and the challenging fabrication (“Integration of the self-similar trampolines with a backside window is possible but complicates the fabrication process; for this reason, in the device displayed in Fig. 4a, Si substrate is still present below the trampoline”).

3. The authors claim they have matched the stress for all beams (constant tension throughout the structure to minimize static in-plane deformations upon structure undercut). Does this help maximizing Q ? If not, I recommend to clearly state that this is only related to fabrication.

We now clarify in the text that this choice is related to fabrication constraints and control of the buckling instability, which is prominent due to the thin (20 nm) Si_3N_4 film.

Amendments to the manuscript

We attempted to improve the clarity of this sentence: “The widths of the segments are determined by requiring constant stress throughout the structure, which minimizes static in-plane deformations upon structure undercut for higher fabrication yield.”

4. In Fig. 2d the authors mentioned that gas damping could be the dominant source for the difference between experiment and simulation. I recommend to also mention that gas damping is higher at a lower frequency. In addition, the conclusion section mentions that it would be possible to exceed a billion Q in room temperature if suppressing other loss mechanisms. It would be relevant to explain how this can be achieved, for example can gas damping be reasonably decreased with improved vacuum?

We thank the reviewer for this suggestion and did consider the impact of gas damping on the measured damping rates. However, we were able to rule out gas damping by performing our ringdown measurements at ultra-high vacuum levels ($P \sim 10^{-9}$ mbar).

We agree with the reviewer regarding the clarification on the gas damping limits for low frequencies. In the free molecular flow regime the quality factor limit due to surrounding gas is given by [2]:

$$Q_{\text{gas}} = \frac{\rho \omega h}{\Gamma_g P} \approx 4.2 \cdot 10^8 \cdot f [\text{MHz}] \cdot \frac{h/(20 [\text{nm}])}{P/(10^{-6} [\text{mbar}])} \cdot \sqrt{T/(300 [\text{K}])}$$

where $\Gamma_g \propto \sqrt{\frac{m}{2k_B T}}$ is the damping rate due to collision with gas molecules of average mass m . For a 20nm-thick resonator at 57 kHz at $P = 5 \times 10^{-9}$ mbar results in $Q_{\text{gas}} \approx 10^{10}$, ruling out gas damping as a limiting factor in the Q measurements of our devices.

We speculate that the additional losses could be caused by residual acoustic wave radiation to the chip bulk and fabrication imperfections caused by wide branching segments.

Amendments to the manuscript

We added the following information to the manuscript: “One potential extrinsic loss mechanism - damping by residual gas in the vacuum chamber - was experimentally ruled out at room temperature ($Q_{\text{gas}} \approx 10^{10}$ for a 20 nm-thick resonator at 57 kHz at $P = 5 \times 10^{-9}$ mbar [2])”

5. Also in Fig. 2d, the 0.7 mm resonator at high frequency exhibits a strong mismatch between simulation and experiment. What is the reason for this?

We do not have a concrete explanation for why there is a mismatch at higher frequencies, but we speculate that this could be due to mode coupling of the high order modes with other, lossier modes of the structure (in-plane or torsional modes) or fabrication imperfections.

6. I recommend to add more details about the finite element model, such as the mesh size and the simulation time, among others. Even if some of these details can be found in reference [9], at least be explicit about it.

Amendments to the manuscript

We added a section in the methods providing more details on the finite element simulation parameters.

7. In the methods section, the intrinsic Q value is mentioned to be two times higher than the average value present in other literature. Why is this the case?

We obtain the Si_3N_4 films for this work from an external provider (Hahn-Schickard Gesellschaft für angewandte Forschung e.V.) and their LPCVD process appears to result in a higher intrinsic quality factor than that from other sources, such as our own cleanroom. The supplier of Si_3N_4 is mentioned in the Methods section.

8. It might be relevant to include an additional reference (I am not an author of this reference, neither do I have any connection to the authors) that performed the experimental validation of the topology optimization paper [ref. 39 in the article under review] and that was also published in Nature communications 12, 5766 (2021).

We became aware of this work, submitted to arXiv.org shortly after ours, after publishing the preprint of our manuscript.

Amendments to the manuscript

We added the reference to Høj et al. [4].

Reviewer 2:

In this work, the authors experimentally verify their nanomechanical designs from a previous publication [ref. 9] with established fabrication and ring-down measurements. In doing so, the authors attain Qs of 780 million with a fundamental mode (at 300K) and > 1 billion (at 6k). The paper is thorough in terms of its characterization and theory and expands the use of their designs to membrane structures, which is novel with respect to ideas presented in ref. 9. What is dubious is how this work fits in within the existing state-of-the-art literature and the impact it can have on the community given that there are alternatives in the field that operate in the same regime with similar (if not better) performance. It is also not so clear how well these devices can be realistically implemented into experiments. As Nature Communications aims to publish state-of-the-art research which can make an impact on the field, it would be important for the authors to address the following points before considering for publication:

1. Higher f and Q are ultimately preferred for quantum optomechanics experiments – one the authors' motivations for the hierarchical resonators. If one requires higher frequencies (~1 MHz) and higher Q, one can look towards nanostring resonators published by the same

group (ref. 25, Gadhimi et al.) where the frequency is nearly 10x higher and Qs larger by 100 million than ones shown in this study. The main argument the authors make for a hierarchical resonator (over resonators with phononic bandgap structures) is that hierarchical resonators uses a fundamental mode which has a large frequency spacing between it and next adjacent mode. However, the higher order modes in nanostrings (like those in ref. 25) can attain a similar relative frequency spacing between the mode of interest and the next adjacent mode via the phononic bandgap. While lower order modes conventionally equate to larger frequency spacing, phononic bandgap structures evade that rule by creating a large bandgaps around higher-order modes. It is not so much the mode number that is important, but the spectral spacing which is achievable between the mode of interest and adjacent modes. In this sense ref. 9 does this just as well – but with a higher Q and frequency which is an order of magnitude larger! It seems phononic bandgap resonators are still significantly more desirable (than hierarchical structures) for quantum optomechanics experiments. Can the authors comment on this?

We would like to thank the reviewer for their careful assessment of our work. There are two main advantages in mode spacing when comparing the binary tree beams and phononic crystal (PnC) bandgap engineering: larger relative mode spacing (for similar mode frequency of binary-trees and PnC devices) and larger separation between the high Q out-of-plane (OP) mode and the low Q in-plane (IP) mode. For a matched fundamental mode frequency (~ 1MHz), binary-tree beams have larger OP mode spacing (MHz range) compared to the PnC soft-clamped modes (few hundred kHz). The other advantage is larger mode spacing between the high Q OP mode and other low Q modes such as IP and torsional modes. Ghadimi et al. show that there are several modes very close to the high Q mode inside the bandgap (few kHz away). This is different in binary-tree resonators and the mode separation between the OP mode and other modes is not only larger but also controllable by means of the branching parameters. In some optomechanical platforms, e.g. near field-coupled systems [5]–[7], different mode families can present sizeable optomechanical coupling, hence making a larger frequency separation desirable.

A subtle drawback of PnC devices is that, when integrated in an optomechanical cavity, many mechanical modes will necessarily be strongly coupled to the optical field. This could complicate experiments by generating additional noise in the cavity field [8], [9], or by requiring more elaborate feedback, stabilization and locking schemes [10], hence the citation to Ref. [8] in the manuscript abstract. A soft clamped fundamental mode, therefore, provides a better approximation of an idealized single-mode optomechanical system.

Finally, we would like to note that a considerable advantage of our method is that it can be applied to trampoline membranes, which can be readily integrated into optomechanical membrane-in-the-middle cavities.

Amendments to the manuscript

We added a section in the methods providing a simulation that shows that the mode spacing of our devices is larger than that of soft-clamped structures.

2. If one wants similar “low” frequency resonators (on the order of $\sim 100\text{kHz}$) and higher Q s, the same group of this study have recently published ‘perimeter mode’ nanomechanical resonators, arxiv:2108.03615 (2021) with $Q > 3$ billion at similar frequencies, and another group (Advanced Materials (2021): 2106248) has also demonstrated structures $\sim 100\text{kHz}$ and $Q > 10^9$, both in room temperature. These ‘perimeter mode’ resonators also have similar sizes as hierarchical resonators and higher Q s. Although they do not use fundamental modes, the mode spacing are comparable which is the big reason to target the fundamental mode. Given the similar operating regime it would make sense to discuss in terms of how hierarchical structures fit into the existing state-of-the-art.

We thank the referee for this comment. We would like to point out that the work on perimeter modes and the spider-web design were published after the work was posted on arXiv and submitted to the journal, thus we did not include any comparison with these designs. In addition, in our case the study of hierarchical resonators and their fundamental unit, the three-beam junction, provided the inspiration to conceive the perimeter mode resonators.

We included a paragraph referencing perimeter modes and “spider-web” designs.

Amendments to the manuscript

We added the following sentence to the introduction: “Other designs such as perimeter modes [11] and spider-web resonators [12] have also demonstrated low dissipation at low frequencies, however not for the fundamental mode of the structure.”

3. One of the prominent differences between this study and designs in ref.9 is the use of hierarchical structures to produce trampoline membranes. The main motivation for membrane structures is to interface the central pad with large optical beam paths or optical cavities (in a way that nanostring cannot). Here it should be noted that the steering wheel structure shown in Fig 4b is very similar to topologically optimized resonators which were recently demonstrated in Nat. Communications (12, 5766 (2021)). Can the authors comment on any fundamental differences or similarities?

We thank the referee for pointing out the similarity with the contemporaneous publication [4], which we are now referencing in the manuscript. The design D1 in the aforementioned work appears indeed very similar to the steering wheel membranes we present, despite the pattern being less regular at the scale of tens of micrometers (see Fig. 6 in [4]), probably a natural outcome of the topology optimization algorithm. As we discuss in our paper, we conceived the steering wheel design in a completely different process, by trying to simplify the design of hierarchical trampolines in order to fit them in a rectangular window, thus facilitating fabrication. To our understanding, the main difference in their work is the assumption of both intrinsic and phonon tunneling losses in the resonant mode for the sake of the optimization algorithm, leading to different designs D1-D5 according to the assumed participation ratio between the two loss sources. We did not consider phonon tunneling in our numerical modelling, as it is often unpredictable and dependent on substrate geometry and mounting conditions, and we remark that the authors of [4] found in the end intrinsic loss to be dominant for all their best-performing devices. The highest quality factors achieved in [4] are also generally consistent with the

performance of our best steering wheel devices, accounting for a decrease in Q with frequency which is inherent to dissipation dilution [13].

4. In Fig 4a, the authors show a “self-similar” trampoline with Q that is 60% larger than that shown in the Nat Comms 12, 5766 (2021) and the steering wheel trampoline, but it is only etched 80um into the silicon substrate. Without etching through the substrate, it would not be possible to optically access the pad or interface with the resonator with an optical cavity (i.e. the main purpose of a pad). Is there a reason this was not etched all the way through the substrate? Is the fabrication prohibitively difficult? At the moment it is not discussed at all and would be important for researchers considering to implement themselves. I would image the the edges of self-similar patterning would not line up well with the square windows etched into the silicon by KOH. Would the extra overhang somehow affect the performance? It is important to note these limitations in fabrication since it would indicate whether the resonators are realistic and viable for researchers wanting to implement such structures into optomechanics experiments. As shown, it does not seem like it is currently possible to interface ‘self-similar’ trampolines into high-finesse cavities, making their usefulness limited for such applications.

It is feasible to include a backside window for optical access below the trampoline resonators, as we show in the Figure below for a different device. However, this complicates slightly the fabrication process, as it requires the patterning of a backside Si_3N_4 mask, potentially decreases the yield of the fragile trampoline structures (a through-hole implies more violent liquid flows in a direction perpendicular to the membrane pad during the cleaning and rinsing steps) and requires careful KOH etch timing in order not to create excess overhang at the clamping points. We note that the trampoline membrane is carefully inserted in a frame with convex corners sufficiently far from the tethers, minimizing the extent of the overhang that is created during undercut if the KOH etch is timed correctly.

For these reasons and in order to optimize the contrast of the SEM image displayed in Fig. 4a, the device presented in the main text was not completely undercut.

Amendments to the manuscript:

We have clarified in the main text the possibility of fabricating structures with an optical window, “Integration of the self-similar trampolines with a backside window is possible but complicates the fabrication process; for this reason, in the device displayed the Si substrate is still present below the trampoline”.

5. One of the promises of hierarchical structures from ref. 9, is that with higher number of branchings, N , one can attain higher Q s. In ref. 9, $N = 4, 5$ are investigated and show higher Q s. However, in Fig 2, the author’s experimental progress stops at $N = 3$ without much explanation. This would be very important to comment on since it touches on the a possible limitation of this design in real-life implementation. Adding a discussion related to this would be very interesting for readers.

We thank the reviewer for this observation. The device characterized in Fig. 3 features $N = 4$ and $Q \approx 3 \cdot 10^8$ in agreement with the numerical predictions. This particular device had a branching angle $\theta = 60^\circ$, therefore allowing a stress-preserving tree with identical branch widths [1]. Devices with larger $\theta > 75^\circ$ (corresponding to higher dissipation dilution) and many branching levels become difficult to fabricate, due to the exponential increase of tether width in stress-preserving structures, $w_N \propto \left(\frac{1}{2 \cos \theta}\right)^N$. Such tethers require long durations of KOH undercut, which can be comparable to the time required to fully undercut the structure supports (pads).

Moreover, with increased numbers of branching levels, it becomes more challenging to fit all the required pads without structure self-contact [1]. For this reason, at $\theta > 75^\circ$, binary tree beams are limited by practical considerations to $N \lesssim 4$, unless the fabrication method or design is modified.

Amendments to the manuscript:

We now mention the fabrication constraints limiting the exploration of binary tree beams to $N < 4$ in Fig. 2: “Devices with $N > 3$ and large branching angles, $\theta > 60^\circ$, are challenging to fabricate, because of the segment width growth in higher branching generations, and due to spatial constraints of the pad supports (see Fig. 2c)”.

6. For clarity, I highly recommend matching the numbers describing the quality factor in the abstract. The experimental results in this paper (which is the essence rather than the simulation) show 0.78 billion for the room temperature Q and 1.1 billion for the cryogenic temperature Q. Stating the value more precisely will help the paper be more straightforward for readers. (ex. 7.8×10^8 Q for room temperature, 1.1×10^9 for cryogenic). Mixing in simulated Qs, with experimental numbers is very confusing for readers.

Amendments to the manuscript:

We changed the abstract sentence to “Here, we apply hierarchical design to silicon nitride nanomechanical resonators and realize binary tree-shaped resonators with room temperature quality factors as high as 7.8×10^8 at 107 kHz frequency (1.1×10^9 at $T = 6$ K), reaching the parameter regime of levitated particles [10, 11]”.

To conclude: On its own, the authors' paper presents concepts which are very interesting and well organized. Looking at the broader literature, my impression is that there are existing systems which could operate with similar (and sometimes better) performance at high Q & f, or , low f & high Q and large mode spacing. The hierarchical designs are definitely creative but not new given ref. 9 - this leads one to assume that the advance must be in the fabrication or characterization techniques but these seem identical to ref. 25 (from the same group). If there is an advance here, the authors have not highlighted this. The question is whether putting them together signifies a significant leap in research. I would say yes if the devices represent state-of-the-art that is not achievable by other systems. This is the highest Q achieved for a fundamental mode, but the mode spacing achieved by this is comparable to other systems which do not use a fundamental mode. As presented it is not clear what is the significant leap that would merit publication in Nat. Comms or whether hierarchical resonators fill a gap that is not possible with existing systems. Addressing the points above could help to clarify this and improve the paper's significance.

We thank the referee for their careful assessment of our work and hope that our answers above have helped to alleviate their concerns.

Reviewer 3:

The authors have addressed the concerns expressed by the reviewers in a fully satisfactory manner.

I only noted the following detail: in their reply letter, the authors suggest to change the wording in the conclusion to "The high stress in our structures ..." whereas in the revised manuscript it still reads "The high dissipation dilution in our structures ...". This should be fixed.

Then I think this will be an excellent contribution to Nature Comms.

We would like to thank the reviewer for their positive evaluation of our work and for their careful reading of the manuscript.

Amendments to the manuscript:

We have fixed the issue and changed the sentence in conclusion from "The high dissipation dilution in our structures..." to "The high stress in our structures..."

References

- [1] S. A. Fedorov, A. Beccari, N. J. Engelsen, and T. J. Kippenberg, "Fractal-like Mechanical Resonators with a Soft-Clamped Fundamental Mode," *Phys. Rev. Lett.*, vol. 124, no. 2, p. 025502, Jan. 2020, doi: 10.1103/PhysRevLett.124.025502.
- [2] M. J. Martin, B. H. Houston, J. W. Baldwin, and M. K. Zalalutdinov, "Damping Models for Microcantilevers, Bridges, and Torsional Resonators in the Free-Molecular-Flow Regime," *J. Microelectromechanical Syst.*, vol. 17, no. 2, pp. 503–511, Apr. 2008, doi: 10.1109/JMEMS.2008.916321.
- [3] P.-L. Yu *et al.*, "A phononic bandgap shield for high- Q membrane microresonators," *Appl. Phys. Lett.*, vol. 104, no. 2, p. 023510, Jan. 2014, doi: 10.1063/1.4862031.
- [4] D. Høj, F. Wang, W. Gao, U. B. Hoff, O. Sigmund, and U. L. Andersen, "Ultra-coherent nanomechanical resonators based on inverse design," *Nat. Commun.*, vol. 12, no. 1, p. 5766, Oct. 2021, doi: 10.1038/s41467-021-26102-4.
- [5] R. Schilling, H. Schütz, A. H. Ghadimi, V. Sudhir, D. J. Wilson, and T. J. Kippenberg, "Near-Field Integration of a SiN Nanobeam and a $\{\mathrm{SiO}\}_2$ Microcavity for Heisenberg-Limited Displacement Sensing," *Phys. Rev. Appl.*, vol. 5, no. 5, p. 054019, May 2016, doi: 10.1103/PhysRevApplied.5.054019.
- [6] V. Sudhir *et al.*, "Appearance and Disappearance of Quantum Correlations in Measurement-Based Feedback Control of a Mechanical Oscillator," *Phys. Rev. X*, vol. 7, no. 1, p. 011001, Jan. 2017, doi: 10.1103/PhysRevX.7.011001.
- [7] B. D. Hauer, P. H. Kim, C. Doolin, F. Souris, and J. P. Davis, "Two-level system damping in a quasi-one-dimensional optomechanical resonator," *Phys. Rev. B*, vol. 98, no. 21, p. 214303, Dec. 2018, doi: 10.1103/PhysRevB.98.214303.
- [8] S. A. Fedorov, A. Beccari, A. Arabmoheghi, D. J. Wilson, N. J. Engelsen, and T. J. Kippenberg, "Thermal intermodulation noise in cavity-based measurements," *Optica*, vol. 7, no. 11, pp. 1609–1616, Nov. 2020, doi: 10.1364/OPTICA.402449.

- [9] J. Guo, R. Norte, and S. Gröblacher, “Feedback Cooling of a Room Temperature Mechanical Oscillator close to its Motional Ground State,” *Phys. Rev. Lett.*, vol. 123, no. 22, p. 223602, Nov. 2019, doi: 10.1103/PhysRevLett.123.223602.
- [10] M. Rossi, D. Mason, J. Chen, Y. Tsaturyan, and A. Schliesser, “Measurement-based quantum control of mechanical motion,” *Nature*, vol. 563, no. 7729, pp. 53–58, Nov. 2018, doi: 10.1038/s41586-018-0643-8.
- [11] M. J. Bereyhi *et al.*, “Nanomechanical resonators with ultra-high-Q perimeter modes,” *ArXiv210803615 Cond-Mat Physicsphysics Physicsquant-Ph*, Dec. 2021, Accessed: Jan. 11, 2022. [Online]. Available: <http://arxiv.org/abs/2108.03615>
- [12] D. Shin, A. Cupertino, M. H. J. de Jong, P. G. Steeneken, M. A. Bessa, and R. A. Norte, “Spiderweb Nanomechanical Resonators via Bayesian Optimization: Inspired by Nature and Guided by Machine Learning,” *Adv. Mater.*, vol. n/a, no. n/a, p. 2106248, Oct. 2021, doi: 10.1002/adma.202106248.
- [13] S. A. Fedorov *et al.*, “Generalized dissipation dilution in strained mechanical resonators,” *Phys. Rev. B*, vol. 99, no. 5, p. 054107, Feb. 2019, doi: 10.1103/PhysRevB.99.054107.

REVIEWER COMMENTS

Reviewer #1 (Remarks to the Author):

The authors have addressed all the points raised by me. I recommend the article for publication in this journal.

Reviewer #2 (Remarks to the Author):

Reviewer's Latest comments written after asterisks (*).

* The author's provide comprehensive answers to the previous questions although some points have been overlooked which would be interesting for readers.

2. If one wants similar “low” frequency resonators (on the order of $\sim 100\text{kHz}$) and higher Qs, the same group of this study have recently published ‘perimeter mode’ nanomechanical resonators, arxiv:2108.03615 (2021) with $Q > 3$ billion at similar frequencies, and another group (Advanced Materials (2021): 2106248) has also demonstrated structures $\sim 100\text{kHz}$ and $Q > 10^9$, both in room temperature. These ‘perimeter mode’ resonators also have similar sizes as hierarchical resonators and higher Qs. Although they do not use fundamental modes, the mode spacing are comparable which is the big reason to target the fundamental mode. Given the similar operating regime it would make sense to discuss in terms of how hierarchical structures fit into the existing state-of-the-art.

Authors' Response:

Amendments to the manuscript

We added the following sentence to the introduction: “Other designs such as perimeter modes [11] and spider-web resonators [12] have also demonstrated low dissipation at low frequencies, however not for the fundamental mode of the structure.”

*It is good that the authors address these works. However, the question remains -- the work [11] & [12] clearly demonstrate that large mode spacing and significantly lower dissipation can be achieved without

a fundamental mode. Given this, why should the reader care about fundamental modes? Making this more clear would improve the main motivation of the paper.

4. In Fig 4a, the authors show a “self-similar” trampoline with Q that is 60% larger than that shown in the Nat Comms 12, 5766 (2021) and the steering wheel trampoline, but it is only etched 80um into the silicon substrate. Without etching through the substrate, it would not be possible to optically access the pad or interface with the resonator with an optical cavity (i.e. the main purpose of a pad). Is there a reason this was not etched all the way through the substrate? Is the fabrication prohibitively difficult? At the moment it is not discussed at all and would be important for researchers considering to implement themselves. I would imagine the the edges of self-similar patterning would not line up well with the square windows etched into the silicon by KOH. Would the extra overhang somehow affect the performance? It is important to note these limitations in fabrication since it would indicate whether the resonators are realistic and viable for researchers wanting to implement such structures into optomechanics experiments. As shown, it does not seem like it is currently possible to interface ‘self-similar’ trampolines into high-finesse cavities, making their usefulness limited for such applications.

Authors' Response: It is feasible to include a backside window for optical access below the trampoline resonators, as we show in the Figure below for a different device. However, this complicates slightly the fabrication process, as it requires the patterning of a backside Si₃N₄ mask, potentially decreases the yield of the fragile trampoline structures (a through-hole implies more violent liquid flows in a direction perpendicular to the membrane pad during the cleaning and rinsing steps) and requires careful KOH etch timing in order not to create excess overhang at the clamping points. We note that the trampoline membrane is carefully inserted in a frame with convex corners sufficiently far from the tethers, minimizing the extent of the overhang that is created during undercut if the KOH etch is timed correctly. For these reasons and in order to optimize the contrast of the SEM image displayed in Fig. 4a, the device presented in the main text was not completely undercut.

*It is great to see that the authors have fabricated a device which is fully etched through so that optical access is possible for realistic optomechanics experiments. Assuming this is best image a fully etched through device, what is very noticeable is the damage and contamination caused by the extra processing required (see attached annotated image)

*It seems that it not only “complicates slightly the fabrication” but genuinely makes it hard to get a good quality sample. And this is important in terms of the design’s feasibility for other researchers. So the simple question is: what is the best quality factor you achieved with a fully-etched “self-similar” trampoline? Do their performance differ significantly from those which are only etched 80um into the substrate? If so, it is important to be clear about this since it is crucial for readers who consider these designs for “interferometric position measurements in Fabry-Perot cavities” (as stated in the abstract).

5. *The authors mention amendments to the manuscript:

"We now mention the fabrication constraints limiting the exploration of binary tree beams to $N < 4$ in Fig. 2: "Devices with $N > 3$ and large branching angles, $\theta > 60^\circ$, are challenging to fabricate, because of the segment width growth in higher branching generations, and due to spatial constraints of the pad supports (see Fig. 2c)".

*I do not see where this was included. Please do.

*In general, I am satisfied with the authors' careful consideration of points brought up. It seems the authors' amendments to address concern #5 were simply forgotten. It is important for the authors to answer remaining questions on concerns #1 & #4 since they are tied to the main motivations of their manuscript.

We would like to thank the reviewers for their consideration and detailed evaluation of our manuscript. In the following sections we address the comments and questions raised by the reviewer (blue sections) and describe the amendments to the manuscript (red sections).

Reviewer 1:

The authors have addressed all the points raised by me. I recommend the article for publication in this journal.

We would like to thank the reviewer for their positive evaluation of our work.

Reviewer 2:

The author's provide comprehensive answers to the previous questions although some points have been overlooked which would be interesting for readers.

We would like to thank the reviewer for continuing the scientific discussion and carefully checking the proposed manuscript changes. In the next paragraphs, we provide point-by-point answers and describe the amendments made to the manuscript.

If one wants similar “low” frequency resonators (on the order of ~100kHz) and higher Qs, the same group of this study have recently published ‘perimeter mode’ nanomechanical resonators, arxiv:2108.03615 (2021) with $Q > 3$ billion at similar frequencies, and another group (Advanced Materials (2021): 2106248) has also demonstrated structures ~100KHz and $Q > 10^9$, both in room temperature. These ‘perimeter mode’ resonators also have similar sizes as hierarchical resonators and higher Qs. Although they do not use fundamental modes, the mode spacing are comparable which is the big reason to target the fundamental mode. Given the similar operating regime it would make sense to discuss in terms of how hierarchical structures fit into the existing state-of-the-art.

Authors' Response:

Amendments to the manuscript

We added the following sentence to the introduction: “Other designs such as perimeter modes [11] and spider-web resonators [12] have also demonstrated low dissipation at low frequencies, however not for the fundamental mode of the structure.”

*It is good that the authors address these works. However, the question remains -- the work [11] & [12] clearly demonstrate that large mode spacing and significantly lower dissipation can be achieved without a fundamental mode. Given this, why should the reader care about fundamental modes? Making this clearer would improve the main motivation of the paper.

We thank the reviewer for this opportunity to further clarify the advantages of our resonators. While the reviewer is correct that the dissipation rates presented in [1], [2] are a factor of 4 and 2 lower respectively, it is not correct that they achieve as large mode spacing as in hierarchical resonators. A crucial point is that the in-plane mode of hierarchical resonators has greater frequency spacing than in alternative designs, including polygon resonators, where the in-plane

mode is in fact very close to the out-of-plane perimeter mode. We note that in Fig 6C of [2] a spectrum is shown showing adjacent modes at 120 kHz for a ‘spiderweb’ mode of 133.6 kHz, a relative mode spacing of about $\frac{\delta f}{f} \approx 10\%$ for the nearest out-of-plane mode (OP) (presumably, as this is not discussed in the paper), compared to the $\frac{\delta f}{f} \approx 40\%$ observed for OP modes in hierarchical resonators. There also appears to be a mode very near the spiderweb mode indicated by the red arrow below, though this cannot be conclusively shown without a higher resolution spectrum.

This would be in line with our expectations from our own work, showing below the thermomechanical spectrum of a polygon resonator (unpublished) with a perimeter mode frequency around 1.08 MHz showing multiple in-plane modes in close vicinity of the perimeter mode.

Furthermore, the only alternative design that can be applied to membranes is phononic crystal patterning. Our best hierarchical membrane shows a thermal-limited force sensitivity of $3.7 \text{ aN}/\sqrt{\text{Hz}}$, beating the state of the art by a factor of three [3]–[5].

For hierarchical resonators, engineering of the fundamental mode also ensures that the mode of interest has the lowest dissipation rate of all the modes of the resonator. In optomechanics experiments, it is often seen that other high- Q modes of the resonator become unstable at high powers and must be stabilized. This can usually be done by, for example, implementing a more sophisticated feedback scheme [6], [7], but is not always a trivial task.

Despite these advantages, we realize that the optimal resonator design will always be application-specific. We consider hierarchical resonators a valuable and meaningful addition to the range of existing dissipation dilution engineering techniques with unique advantages compared to other resonator classes. Our paper provides a detailed description of the novelty, functionality and performance of our devices and provides enough detail for the reader to evaluate this resonator design for their application.

As a side note, we would like to clarify the chronology of the work. This work on hierarchical resonators appeared on arXiv March 2021 [8] and in fact predates these two other works by five months [1], [2]. In our group, the polygon resonators were conceived following the work presented in this paper and have a strong conceptual dependence on the device designs explored here.

Amendments to the manuscript:

We have attempted to clarify the advantages of our resonators by adding a sentence in the introduction about the frequency separation of our resonators:

“Compared to other soft-clamped resonators, hierarchical resonators offer the largest relative frequency separation for the high- Q mode (see Methods).”

We have also moved the sentence about perimeter modes and spiderweb resonators earlier in the manuscript, which now reads:

“Contemporaneous work such as perimeter modes and “spider-web” resonators have demonstrated low dissipation at low frequencies, but not for the fundamental mode of the structure”

In Fig 4a, the authors show a “self-similar” trampoline with Q that is 60% larger than that shown in the Nat Comms 12, 5766 (2021) and the steering wheel trampoline, but it is only etched 80um into the silicon substrate. Without etching through the substrate, it would not be possible to optically access the pad or interface with the resonator with an optical cavity (i.e. the main purpose of a pad). Is there a reason this was not etched all the way through the substrate? Is the fabrication prohibitively difficult? At the moment it is not discussed at all and would be important for researchers considering to implement themselves. I would image the the edges of self-similar patterning would not line up well with the square windows etched into the silicon by

KOH. Would the extra overhang somehow affect the performance? It is important to note these limitations in fabrication since it would indicate whether the resonators are realistic and viable for researchers wanting to implement such structures into optomechanics experiments. As shown, it does not seem like it is currently possible to interface ‘self-similar’ trampolines into high-finesse cavities, making their usefulness limited for such applications.

Authors' Response: It is feasible to include a backside window for optical access below the trampoline resonators, as we show in the Figure below for a different device. However, this complicates slightly the fabrication process, as it requires the patterning of a backside Si₃N₄ mask, potentially decreases the yield of the fragile trampoline structures (a through-hole implies more violent liquid flows in a direction perpendicular to the membrane pad during the cleaning and rinsing steps) and requires careful KOH etch timing in order not to create excess overhang at the clamping points. We note that the trampoline membrane is carefully inserted in a frame with convex corners sufficiently far from the tethers, minimizing the extent of the overhang that is created during undercut if the KOH etch is timed correctly. For these reasons and in order to optimize the contrast of the SEM image displayed in Fig. 4a, the device presented in the main text was not completely undercut.

*It is great to see that the authors have fabricated a device which is fully etched through so that optical access is possible for realistic optomechanics experiments. Assuming this is best image a fully etched through device, what is very noticeable is the damage and contamination caused by the extra processing required (see attached annotated image)

*It seems that it not only “complicates slightly the fabrication” but genuinely makes it hard to get a good quality sample. And this is important in terms of the design’s feasibility for other researchers. So the simple question is: what is the best quality factor you achieved with a fully-etched “self-similar” trampoline? Do their performance differ significantly from those which are only etched 80um into the substrate? If so, it is important to be clear about this since it is crucial for readers who consider these designs for “interferometric position measurements in Fabry-Perot cavities” (as stated in the abstract).

We thank the reviewer for this observation. For fully undercut self-similar trampolines, we observed quality factors up to $\approx 10^7$, but with a different design that we conceived in the early stages of the project and showed buckling instability issues (see corresponding Methods section). Regarding the new, amended design that we present in the manuscript and we will include in the data repository, we have only fabricated membranes with partial undercut in the substrate, but, based on our previous experience, we predict that it should be possible to realize devices with a backside window. The presence of a backside window does not impose design changes nor increase the possibility of device damage (in fact, the trampoline devices we demonstrated were processed on a wafer which also contained backside-released samples, on which the extra steps necessary to create a backside window were carried out), but might decrease the final yield of suspended devices.

The SEM image we attached in our previous reply displayed a device with anomalous contamination on the tethers, that we purposefully did not characterize and kept only for imaging and illustration purposes.

Amendments to the manuscript:

We amended the manuscript paragraph as follows: “Integration of the self-similar trampolines with a backside window is possible but complicates the fabrication process; for this reason, in the device displayed in 4a, the Si substrate is still present below the trampoline. We were able to fabricate devices with a backside window, but they employed an earlier design exhibiting buckling upon device release (see Methods section), therefore they showed lower dissipation dilution.”.

*The authors mention amendments to the manuscript:

"We now mention the fabrication constraints limiting the exploration of binary tree beams to $N < 4$ in Fig. 2: “Devices with $N > 3$ and large branching angles, $\theta > 60^\circ$, are challenging to fabricate, because of the segment width growth in higher branching generations, and due to spatial constraints of the pad supports (see Fig. 2c)”.

*I do not see where this was included. Please do.

Amendments to the manuscript:

We would like to apologize for this mistake. This statement is now added to the revised manuscript, in the section “Binary tree resonators”.

*In general, I am satisfied with the authors’ careful consideration of points brought up. It seems the authors’ amendments to address concern #5 were simply forgotten. It is important for the authors to answer remaining questions on concerns #1 & #4 since they are tied to the main motivations of their manuscript.

References

- [1] M. J. Breyhi *et al.*, “Nanomechanical resonators with ultra-high-Q perimeter modes,” *arXiv:2108.03615 [cond-mat, physics:physics, physics:quant-ph]*, Dec. 2021, Accessed: Jan. 27, 2022. [Online]. Available: <http://arxiv.org/abs/2108.03615>
- [2] D. Shin, A. Cupertino, M. H. J. de Jong, P. G. Steeneken, M. A. Bessa, and R. A. Norte, “Spiderweb nanomechanical resonators via Bayesian optimization: inspired by nature and guided by machine learning,” *Advanced Materials*, vol. 34, no. 3, p. 2106248, Jan. 2022, doi: 10.1002/adma.202106248.
- [3] S. A. Fedorov, A. Beccari, A. Arabmoheghi, D. J. Wilson, N. J. Engelsen, and T. J. Kippenberg, “Thermal intermodulation noise in cavity-based measurements,” *Optica, OPTICA*, vol. 7, no. 11, pp. 1609–1616, Nov. 2020, doi: 10.1364/OPTICA.402449.
- [4] C. Reinhardt, T. Müller, A. Bourassa, and J. C. Sankey, “Ultralow-Noise SiN Trampoline Resonators for Sensing and Optomechanics,” *Phys. Rev. X*, vol. 6, no. 2, p. 021001, Apr. 2016, doi: 10.1103/PhysRevX.6.021001.
- [5] I. Galinskiy, Y. Tsaturyan, Y. Tsaturyan, M. Parniak, and E. S. Polzik, “Phonon counting thermometry of an ultracoherent membrane resonator near its motional ground state,” *Optica, OPTICA*, vol. 7, no. 6, pp. 718–725, Jun. 2020, doi: 10.1364/OPTICA.390939.

- [6] M. Rossi, D. Mason, J. Chen, Y. Tsaturyan, and A. Schliesser, “Measurement-based quantum control of mechanical motion,” *Nature*, vol. 563, no. 7729, pp. 53–58, Nov. 2018, doi: 10.1038/s41586-018-0643-8.
- [7] J. Guo, R. Norte, and S. Gröblacher, “Feedback Cooling of a Room Temperature Mechanical Oscillator close to its Motional Ground State,” *Phys. Rev. Lett.*, vol. 123, no. 22, p. 223602, Nov. 2019, doi: 10.1103/PhysRevLett.123.223602.
- [8] A. Beccari *et al.*, “Hierarchical tensile structures with ultralow mechanical dissipation,” *arXiv:2103.09785 [cond-mat, physics:physics, physics:quant-ph]*, Mar. 2021, Accessed: Apr. 19, 2021. [Online]. Available: <http://arxiv.org/abs/2103.09785>

REVIEWER COMMENTS

Reviewer #2 (Remarks to the Author):

The authors' work is creative and thorough and I think it can be published in Nature Communications if they are more direct with the limitations of their system. I appreciated the authors' detailed response, but I missed the same level of detail in the manuscript itself. As a platform published in Nat. Comms. many readers will try to replicate and implement these designs for their own experiments, and perhaps a more direct approach and open descriptions of limitations is lacking – specifically in the case of self-similar trampolines which I think are the highlight of this manuscript.

The self-similar trampolines (Fig 4a) are the most original design in this paper, with the idea being that the hierarchical structure can uniquely accommodate the inclusion of a membrane pad. This pad enables these mechanical resonators to couple with an free-space optical cavity field, which really opens up its uses in optomechanical experiments. In my view, this is what makes their new designs and findings very interesting for broad audiences. However, one needs a hole through the chip to get optical access (and couple with free-space cavities). When the authors say they back-etched into devices, and they've only measured Qs *up to* 10 million and not 100 million they claim, this should be explained in detail and the data shown. The wording "up to 10^7 " insinuates that 10 million is the highest but the average is usually below that ($\sim 10^6$)? If so, that's a large reduction of \sim couple orders of magnitude when back-etching is introduced in the chip. This is much more than just a "lower dissipation dilution" as amended in the manuscript.

For publication, one cannot claim that with updated design, all should work the same without evidence. It needs to be shown with data. If it is prohibitively difficult to demonstrate this high Q (i.e. $\sim 10^8$) with a back-etch for optical access, this should be written in an open and detailed way. This will allow Nat. Comms readers to realistically consider these designs in their own experiments (both the benefits and limitations) and I think it would improve the manuscript and spur more interest.

In summary, the paper is good, the ideas are unique, so no need to downplay limitations with careful language. Being more open with the limitations of the system will make the results more accessible to researchers and will broaden its impact. I highly encourage the authors to include Q data of self-similar trampoline devices with and without back-etching, including discussion of the any differences there may be. In my view, this should be addressed in a genuinely open and detailed way before publishing in Nat. Communications.

The authors' work is creative and thorough and I think it can be published in Nature Communications if they are more direct with the limitations of their system. I appreciated the authors' detailed response, but I missed the same level of detail in the manuscript itself. As a platform published in Nat. Comms. many readers will try to replicate and implement these designs for their own experiments, and perhaps a more direct approach and open descriptions of limitations is lacking – specifically in the case of self-similar trampolines which I think are the highlight of this manuscript.

The self-similar trampolines (Fig 4a) are the most original design in this paper, with the idea being that the hierarchical structure can uniquely accommodate the inclusion of a membrane pad. This pad enables these mechanical resonators to couple with an free-space optical cavity field, which really opens up its uses in optomechanical experiments. In my view, this is what makes their new designs and findings very interesting for broad audiences. However, one needs a hole through the chip to get optical access (and couple with free-space cavities). When the authors say they back-etched into devices, and they've only measured Q_s *up to* 10 million and not 100 million they claim, this should be explained in detail and the data shown. The wording “up to 10^7 ” insinuates that 10 million is the highest but the average is usually below that ($\sim 10^6$)? If so, that's a large reduction of \sim couple orders of magnitude when back-etching is introduced in the chip. This is much more than just a “lower dissipation dilution” as amended in the manuscript.

For publication, one cannot claim that with updated design, all should work the same without evidence. It needs to be shown with data. If it is prohibitively difficult to demonstrate this high Q (i.e. $\sim 10^8$) with a back-etch for optical access, this should be written in an open and detailed way. This will allow Nat. Comms readers to realistically consider these designs in their own experiments (both the benefits and limitations) and I think it would improve the manuscript and spur more interest.

In summary, the paper is good, the ideas are unique, so no need to downplay limitations with careful language. Being more open with the limitations of the system will make the results more accessible to researchers and will broaden its impact. I highly encourage the authors to include Q data of self-similar trampoline devices with and without back-etching, including discussion of the any differences there may be. In my view, this should be addressed in a genuinely open and detailed way before publishing in Nat. Communications.

We thank the reviewer for the overall positive assessment of our paper and for the suggestions to make the conclusions clearer. It was certainly not our intention to downplay the limitation of our design, but we understand that the phrasing of the sentence in the manuscript could be misleading without detailed technical knowledge. We have now included additional information on the inclusion of a backside window in a dedicated section in the Supplementary Information, reporting the mechanical quality factor of the buckled device we referred to in our previous response (the previously-reported $Q \approx 9 \cdot 10^6$ corresponds to the characterization of that single device; we apologize to the reviewer for the confusion). We would like to again note that this reduction in quality factor stems from the buckling of the device¹⁻⁴, and is not due to any contamination from the undercut process itself. This can be evidenced by the fact that we do demonstrate high quality factor devices (>200 million) with the steering wheel design using the same undercut process. We would also like to note that the reported quality factors of the 'steering wheel' design are as high as with the self-similar design. We hope it is now clear in the manuscript that, while the self-similar membrane design does have a higher potential, we have so far not been able to fully harness this potential advantage. As we never attempted to fabricate membranes with the specific design shown in Fig. 4a with a backside window, we cannot comment quantitatively on the yield of such a process.

Pertaining to the reviewer's comments about the use of our devices in optomechanics experiments, we would like to note that in the time that the paper has been under review, the Gröblacher group in Delft has demonstrated an optomechanical system using a mechanical resonator design highly inspired by our binary tree beam design ⁵, which can exploit the spectral purity and high coupling rates that operation with the fundamental mode allows. We therefore believe that not only the membranes, but also the binary tree beam designs will also find application in optomechanical systems.

Amendments to the manuscript

We added a new section in the Supplementary Information with more details about the possibility of opening a backside window, including an optical image of a self-similar trampoline device with undercut. We added a remark in the main text, stating that we could not fully achieve the predicted quality factor with self-similar trampolines: "We remark that the self-similar trampoline membrane exhibited a Q about 3.5 times lower than the finite element prediction, potentially due to out-of-plane static deformations that we observed in the suspended device (see Supplementary information).".

We will furthermore provide all the lithographic masks employed for the fabrication of the samples in this manuscript in a dedicated Zenodo folder, to be released shortly before manuscript publication.

References

1. Ghadimi, A. H. *et al.* Elastic strain engineering for ultralow mechanical dissipation. *Science* **360**, 764–768 (2018).
2. Norte, R. A., Moura, J. P. & Gröblacher, S. Mechanical Resonators for Quantum Optomechanics Experiments at Room Temperature. *Phys. Rev. Lett.* **116**, 147202 (2016).
3. Beccari, A. *et al.* Strained crystalline nanomechanical resonators with ultralow dissipation. *arXiv:2107.02124 [cond-mat, physics:quant-ph]* (2021).
4. Pratt, J. R. *et al.* Nanoscale torsional dissipation dilution for quantum experiments and precision measurement. *arXiv:2112.08350 [cond-mat, physics:physics, physics:quant-ph]* (2021).
5. Guo, J. & Gröblacher, S. Integrated optical-readout of a high-Q mechanical out-of-plane mode. *arXiv:2202.06336 [cond-mat, physics:physics, physics:quant-ph]* (2022).

REVIEWERS' COMMENTS

Reviewer #2 (Remarks to the Author):

The authors have added sufficient explanations and data to the manuscript which clearly highlights their new designs as well as limitations. This makes the research more accessible to the broader audience and I think the manuscript would be appropriate for publication in Nature Communications.